# Cbfβ regulates Wnt/β-catenin, Hippo/Yap, and Tgfβ signaling pathways in articular cartilage homeostasis and protects from ACLT surgery-induced osteoarthritis

Wei Chen[1,2]*, Yun Lu[2], Yan Zhang[2], Jinjin Wu[2], Abigail McVicar[1], Yilin Chen[1], Siyu Zhu[1], Guochun Zhu[2], You Lu[1], Jiayang Zhang[1], Matthew McConnell[1], Yi-Ping Li[1,2]*

[1]Division in Cellular and Molecular Medicine, Department of Pathology and Laboratory Medicine, Tulane University School of Medicine, Tulane University, New Orleans, United States; [2]Department of Pathology, School of Medicine, University of Alabama at Birmingham, Birmingham, United States

*For correspondence:
wchen18@tulane.edu (WC);
yli81@tulane.edu (Y-PingL)

**Abstract** As the most common degenerative joint disease, osteoarthritis (OA) contributes significantly to pain and disability during aging. Several genes of interest involved in articular cartilage damage in OA have been identified. However, the direct causes of OA are poorly understood. Evaluating the public human RNA-seq dataset showed that *CBFB* (subunit of a heterodimeric Cbfβ/Runx1, Runx2, or Runx3 complex) expression is decreased in the cartilage of patients with OA. Here, we found that the chondrocyte-specific deletion of *Cbfb* in tamoxifen-induced *Cbfb*^f/f;*Col2a1-CreER*^T mice caused a spontaneous OA phenotype, worn articular cartilage, increased inflammation, and osteophytes. RNA-sequencing analysis showed that Cbfβ deficiency in articular cartilage resulted in reduced cartilage regeneration, increased canonical Wnt signaling and inflammatory response, and decreased Hippo/Yap signaling and Tgfβ signaling. Immunostaining and western blot validated these RNA-seq analysis results. ACLT surgery-induced OA decreased Cbfβ and Yap expression and increased active β-catenin expression in articular cartilage, while local AAV-mediated *Cbfb* overexpression promoted Yap expression and diminished active β-catenin expression in OA lesions. Remarkably, AAV-mediated *Cbfb* overexpression in knee joints of mice with OA showed the significant protective effect of Cbfβ on articular cartilage in the ACLT OA mouse model. Overall, this study, using loss-of-function and gain-of-function approaches, uncovered that low expression of Cbfβ may be the cause of OA. Moreover, Local admission of *Cbfb* may rescue and protect OA through decreasing Wnt/β-catenin signaling, and increasing Hippo/Yap signaling and Tgfβ/Smad2/3 signaling in OA articular cartilage, indicating that local *Cbfb* overexpression could be an effective strategy for treatment of OA.

## Editor's evaluation

This fundamental work advances our understanding of the role of Cbfβ in maintaining articular cartilage homeostasis and the underlying mechanisms. The evidence supporting the conclusion is convincing. This paper is of potential interest to skeletal biologists and orthopaedic surgeons who study the pathogenesis and the therapeutics of osteoarthritis.

## Introduction

As the most common degenerative joint disease, osteoarthritis (OA) is associated with painful, chronic inflammation that often leads to severe joint pain and joint stiffness for people over the age of 55 (*Shane Anderson and Loeser, 2010*; *Sharma, 2016*). Aging is a major contributor to OA, affecting the knees, hips, and spine and inflicting pain (*Shane Anderson and Loeser, 2010*; *Malfait, 2016*; *Loeser, 2013*; *Aini et al., 2016*). OA is characterized by a multitude of clinical and laboratory findings including osteophyte formation, cartilage degradation, subchondral bone thickening, and elevated cartilage degradation enzymes such as matrix metalloproteinases and aggrecanases (*Sharma, 2016*; *Hunter and Bierma-Zeinstra, 2019*; *Zhen et al., 2013*). Treatment options for joint degeneration in OA are often palliative and oftentimes require surgical interventions such as joint replacement (*Hunter, 2011*), but artificial joints can wear out or come loose and might eventually need to be replaced. As such, a more complete understanding of the mechanisms underlying how transcription factors regulate bone and cartilage formation to maintain bone and cartilage homeostasis could be critical to developing therapies for degenerative joint diseases such as OA.

Recent studies have begun to shed light on the nature of the genetic basis of OA and have confirmed several genes of interest involved in subchondral bone and articular cartilage degeneration including Yap, Sox9, Wnt/β-catenin signaling, and Tgfβ/BMP signaling (*Loeser, 2013*; *Gough, 2011*; *Xia et al., 2014*; *Zhang et al., 2015*; *Karystinou et al., 2015*; *Lane et al., 2017*; *Wu et al., 2012*). Core binding factors are heterodimeric transcription factors consisting of alpha (Cbfα) and beta (Cbfβ) subunits (*Wu et al., 2014a*; *Wu et al., 2014b*). The Cbfβ subunit is a non-DNA-binding protein that binds Cbfα (also known as Runx) proteins to mediate the affinity of their DNA-binding (*Wu et al., 2014a*; *Wu et al., 2014b*). Runx/Cbfβ heterodimers play critical roles in chondrocyte commitment, proliferation, and differentiation, as well as osteoblast differentiation (*Wu et al., 2014a*; *Wu et al., 2014b*; *Westendorf and Hiebert, 1999*; *Tian et al., 2014*; *Chen et al., 2014*; *Park et al., 2016*; *Qin et al., 2015*; *Lim et al., 2015*). Cbfβ was reported as a potential key transcriptional factors in the regulatory network of OA by Gene Expression Omnibus data analysis (*Li et al., 2013*). Yet, the function of Cbfβ in OA pathogenesis remains unclear due to the lack of gain-of-function and loss-of-function animal model studies (*Wu et al., 2014a*). Recently, another study has identified that Cbfβ may play an important role in regeneration and repair of articular cartilage in OA (*Che et al., 2023*). Moreover, a recent study on a small molecule kartogenin showed the crucial role of Cbfβ-Runx1 transcriptional program in chondrocyte differentiation in OA (*Johnson et al., 2012*). However, the underlying mechanism behind Cbfβ regulation in OA remains unclear. In this study, we sought to characterize the mechanisms underlying Cbfβ's regulation in OA and develop potential therapeutic approaches for OA.

In this study, we showed that the deletion of *Cbfb* in the postnatal cartilage in tamoxifen (TMX) induced *Cbfb^{f/f};Col2a1-CreER^T* mice caused a spontaneous OA phenotype, including wear and loss of cartilage, osteophytes, decreased hip joint space, and increased inflammation. Notably, we observed the most severe phenotype in mutant mouse knee joints and hip joints. The loss-of-function study demonstrates the important role of Cbfβ in chondrocyte homeostasis and provides important insights into the role of Cbfβ as a critical transcriptional factor in OA. We also observed that Cbfβ enhanced articular cartilage regeneration and repair by modulating multiple key signaling pathways, including Hippo/Yap, Wnt/β-catenin, Tgfβ, and Sox9. In addition, we demonstrated that adeno-associated virus-mediated local *Cbfb* over-expression protects against surgery-induced OA in mice. The investigation of Cbfβ-multiple signaling regulation helps us better understand the OA genesis mechanism and will potentially facilitate the development of novel treatments for OA.

## Results

### Tamoxifen (TMX) induced *Cbfb^{f/f};Col2a1-CreER^T* developed spontaneous OA

To investigate the role of Cbfβ in spontaneous OA, the expression level of *CBFB* was first examined in human patients with OA by analyzing relevant datasets from published sources (*Fisch et al., 2018*; *Rushton et al., 2014*; *Figure 1A and B*). Interestingly, there was significantly reduced *CBFB* gene expression in cartilage of human OA patients compared to healthy individuals (*Figure 1A*; *Fisch et al., 2018*). Moreover, Methyl-seq data of human OA patient hip tissue exhibited increased methylation at the *CBFB* promoter of OA patients compared to healthy individuals, indicating inhibited *CBFB*

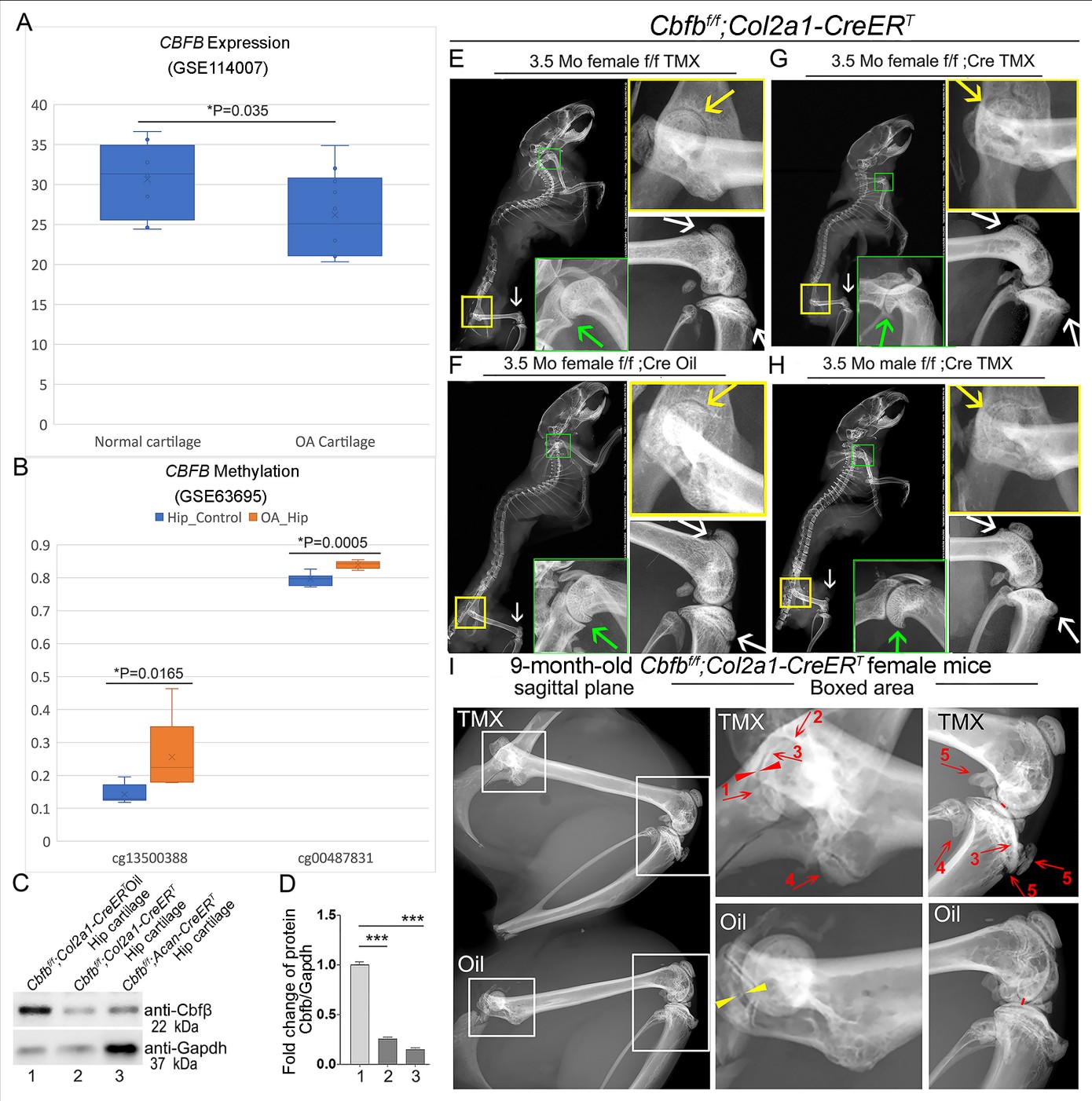

**Figure 1.** Tamoxifen (TMX) induced *Cbfb^(f/f)*;*Col2a1-CreER^T* mice developed spontaneous OA. (**A**) Public human RNA-seq dataset (n=8) (GSE114007) showing *CBFB* mRNA expression level in Normal and OA patient cartilage. (**B**) Public human methyl-seq dataset (n=5) (GSE63695) showing methylation at the *CBFB* promoter region (cg13500388 and cg00487831) in Normal and OA hip tissue. Statistical significance was assessed using Student's t-test. Values were considered statistically significant at p<0.05. (**C**) Western blot to examine Cbfβ protein levels in the hip articular cartilage of 3.5-month-old male oil injected *Cbfb^(f/f)*;*Col2a1-CreER^T* and TMX injected *Cbfb^(f/f)*;*Col2a1-CreER^T*, and 4-month-old male TMX injected *Cbfb^(f/f)*;*Acan-CreER^T* mice (n=3). (**D**) Quantification of (**C**). (**E**) X-ray of 3.5-month-old TMX injected female *Cbfb^(f/f)* mouse hip, shoulder, and knee joint (n=15). (**F**) X-ray of 3.5-month-old oil injected female *Cbfb^(f/f)*;*Col2a1-CreER^T* mouse hip, shoulder, and knee joint (n=15). (**G**) X-ray of 3.5-month-old TMX injected female *Cbfb^(f/f)*;*Col2a1-CreER^T* mouse hip, shoulder, and knee joint (n=12). (**H**) X-ray of 3.5-month-old TMX injected male *Cbfb^(f/f)*;*Col2a1-CreER^T* mouse hip, shoulder, and knee joint. Green arrow: osteophytes in shoulder; yellow arrow: hip joint space; white arrow: hyperosteogeny in knee. (**I**) X-ray image of hips and knee joints of 9-month-old female *Cbfb^(f/f)*;*Col2a1-CreER^T* mice with oil injection and *Cbfb^(f/f)*;*Col2a1-CreER^T* mice with TMX injection (n=9). Red arrow 1,2,3: worn articular cartilage; Red arrow 4,5: osteophytes (spurs); Red arrow head: narrow joint space; Yellow arrow head: healthy hip joint space.

*Figure 1 continued on next page*

Figure 1 continued
The online version of this article includes the following source data for figure 1:
Source data 1. Labeled raw western blot data for *Figure 1C* (anti-Cbfβ and anti-Gapdh).
Source data 2. Unlabeled raw western blot data for *Figure 1C* (anti-Cbfβ and anti-Gapdh).

expression in OA individuals may be through epigenetic regulation (*Figure 1B*; *Rushton et al., 2014*). These data revealed that Cbfβ might play an important role in suppressing OA.

Further, to evaluate the impact of Cbfβ loss-of-function on OA development, TMX inducible *Cbfb^{f/f};Col2a1-CreER^T* and *Cbfb^{f/f};Acan-CreER^T* mice were generated by crossing *Cbfb^{f/f}* mice with either TMX inducible *Col2a1-CreER^T* or *Acan-CreER^T* mouse lines. First, the validity of our mice models was confirmed by western blotting. Cbfβ protein levels were significantly decreased in the hip articular cartilage of both *Cbfb^{f/f};Col2a1-CreER^T* and *Cbfb^{f/f};Acan-CreER^T* mice after TMX injection, indicating successful knockout of *Cbfb* in both mouse models (*Figure 1C and D*). Next, the bone phenotype in *Cbfb* conditional knockout mice was examined. Whole-body X-ray images of 3.5-month-old male and female *Cbfb^{f/f}* and *Cbfb^{f/f};Col2a1-CreER^T* mice after TMX injection showed osteophytes in the shoulder joint compared to *Cbfb^{f/f};Col2a1-CreER^T* mice corn oil or *Cbfb^{f/f}* TMX injection controls (*Figure 1E, G, F and H*, green arrows). X-ray results also revealed that in the TMX-induced *Cbfb^{f/f};Col2a1-CreER^T* mice, the articular cartilage presented unclear borders and narrow hip joint spaces compared to the control groups (*Figure 1E, G, F and H*, yellow arrows). Cbfβ-deficient mice also developed bone hyperosteogeny at the knee joints as shown by X-ray (*Figure 1E, G, F and H*, white arrows). Moreover, TMX injected 9-month-old female *Cbfb^{f/f};Col2a1-CreER^T* mice developed more severe OA phenotypes of joint blurred borders (worn articular cartilage red arrows 1, 2, and 3), osteophytes (bone spurs, red arrows 4 and 5), and narrow joint space (red arrow heads) compared to the oil injected *Cbfb^{f/f};Col2a1-CreER^T* controls (*Figure 1I*, yellow arrow heads indicating healthy hip joint space). These data suggested that Cbfβ-deficient mice develop whole-body bone phenotypes that mimic human OA, and Cbfβ plays an important role in postnatal cartilage regeneration which affects OA onset and progression.

## Deficiency of Cbfβ in cartilage of 3.5-month-old mutant mice resulted in a more severe OA-like phenotype with decreased articular cartilage and osteoblasts, and increased osteoclasts and subchondral bone hyperplasia

To delve deeper into understanding the influence of Cbfβ in regulating the progression of OA, a chronological examination of hip joint histology was conducted, encompassing 1-month-old, 2-month-old, and 3.5-month-old TMX-induced *Cbfb^{f/f};Col2a1-CreER^T* mice. Hematoxylin and eosin (H&E) and Safranin O (SO) staining of 1-month-old *Cbfb^{f/f};Col2a1-CreER^T* mice (2 weeks after TMX induction) hip joints showed thicker femoral head cartilage (*Figure 2A, B and E*, left panel) and slightly decreased tartrate-resistant acid phosphatase (TRAP)-positive cell numbers when compared to the controls (*Figure 2C and F*, left panel). No significant change in alkaline phosphatase (ALP)-positive osteoblast numbers was detected (*Figure 2D and G*, left panel). However, at 2 months old, Cbfβ-deficient mice (6 weeks after TMX induction) hip joints had about twofold cartilage loss in the femoral head (*Figure 2A, B and E*, middle panel) with about twofold increased TRAP-positive osteoclast numbers, indicating increased inflammation (*Figure 2C and F*, middle panel) and threefold decreased ALP-positive osteoblast numbers (*Figure 2D and G*, middle panel). Additionally, a comparable pattern manifested in the hip joints of 3.5-month-old *Cbfb^{f/f};Col2a1-CreER^T* mice (12 weeks after TMX induction). Notably, there were about eightfold decrease in the SO-positive area (*Figure 2A, B and E*, right panel), about 5.5-fold increase in TRAP-positive osteoclasts (*Figure 2C and F*, right panel) and about 10-fold decrease in ALP-positive osteoblasts (*Figure 2D and G*, right panel). It was noticed that there was significant subchondral bone hyperplasia in 3.5-month-old mutant mice (*Figure 2C*, right panel). Collectively, histological data provided additional support, indicating that while Cbfβ did not exert a significant effect on the hip cartilage of 1 month-old mice, deficiency of Cbfβ in cartilage in 3.5-month-old mutant mice resulted in a more severe OA-like phenotype with decreased articular cartilage and osteoblasts, and increased osteoclasts and subchondral bone hyperplasia. Our data

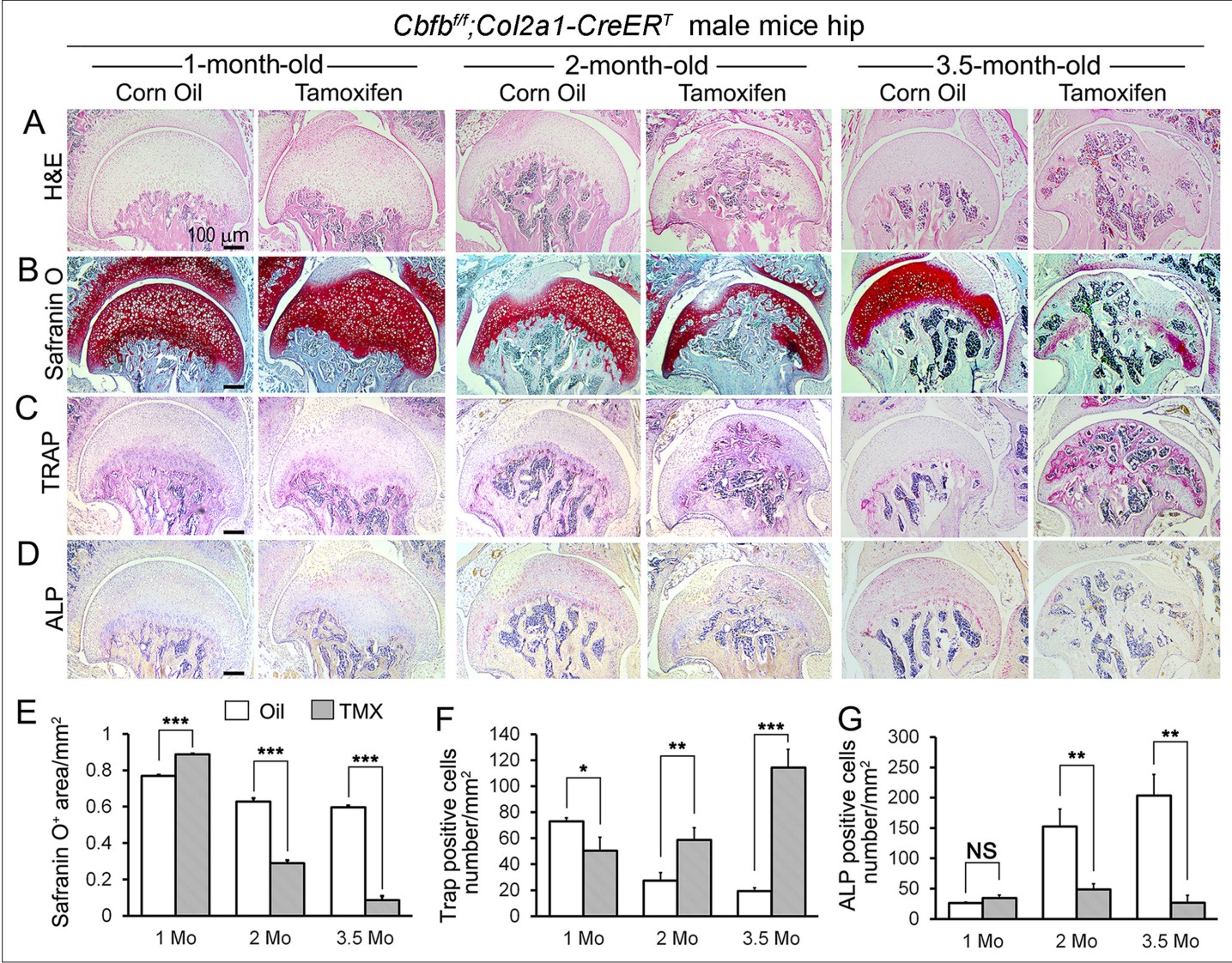

**Figure 2.** *Cbfb* deletion in *Col2a1-CreER^T* mice cartilage resulted in more severe OA-like phenotype 3.5-month-old mutant mice with increased osteoclasts and subchondral bone hyperplasia, decreased articular cartilage and osteoblasts. (**A–D**) H&E staining (**A**), SO staining (**B**), TRAP staining (**C**), and ALP staining (**D**) of 1-month-old, 2-month-old, and 3.5-month-old male *Cbfb^f/f;Col2a1-CreER^T* mice hips respectively. (**E**) Quantification of SO red area of (**B**). Data was measured by ImageJ. (**F**) Quantification of TRAP-positive cell numbers of (**C**). (**G**) Quantification of ALP-positive cell numbers of (**D**). TMX = Tamoxifen, *Cbfb* deleted group; Oil = Corn Oil, control group. n=7. Data are shown as mean ± SD. NS, no significance; *p<0.05; **p<0.01; ***p<0.001 vs. controls by Student's t-test. Scale bar: 100 μm.

further supported that Cbfβ plays a crucial role in articular cartilage regeneration, and deficiency of Cbfβ in mice might lead to the progression of OA.

## The deficiency of Cbfβ may be the cause of early onset OA

Anterior cruciate ligament (ACL) injury is a common cause of human OA, and Anterior cruciate ligament transection (ACLT) is a well-established mouse model that mimics human OA. Bone remodeling between chondrocytes and subchondral bone ossification is known to be important for OA (*Zhen et al., 2013*). In order to further analyze the role of Cbfβ in OA pathological conditions, we developed OA pathological disease mice models by performing ACLT surgery on mice knees. Then, we performed radiographical and histological studies on WT, *Cbfb^f/f*, and *Cbfb^f/f;Col2a1-CreER^T* mice with or without ACLT surgery. We discovered that in *Cbfb^f/f;Col2a1-CreER^T* mice with ACLT surgery, more severe articular cartilage wear (white arrow) showed unclear borders, joint space loss (purple arrow), more hyperosteogeny (blue arrow), and significantly enhanced subchondral bone density (red

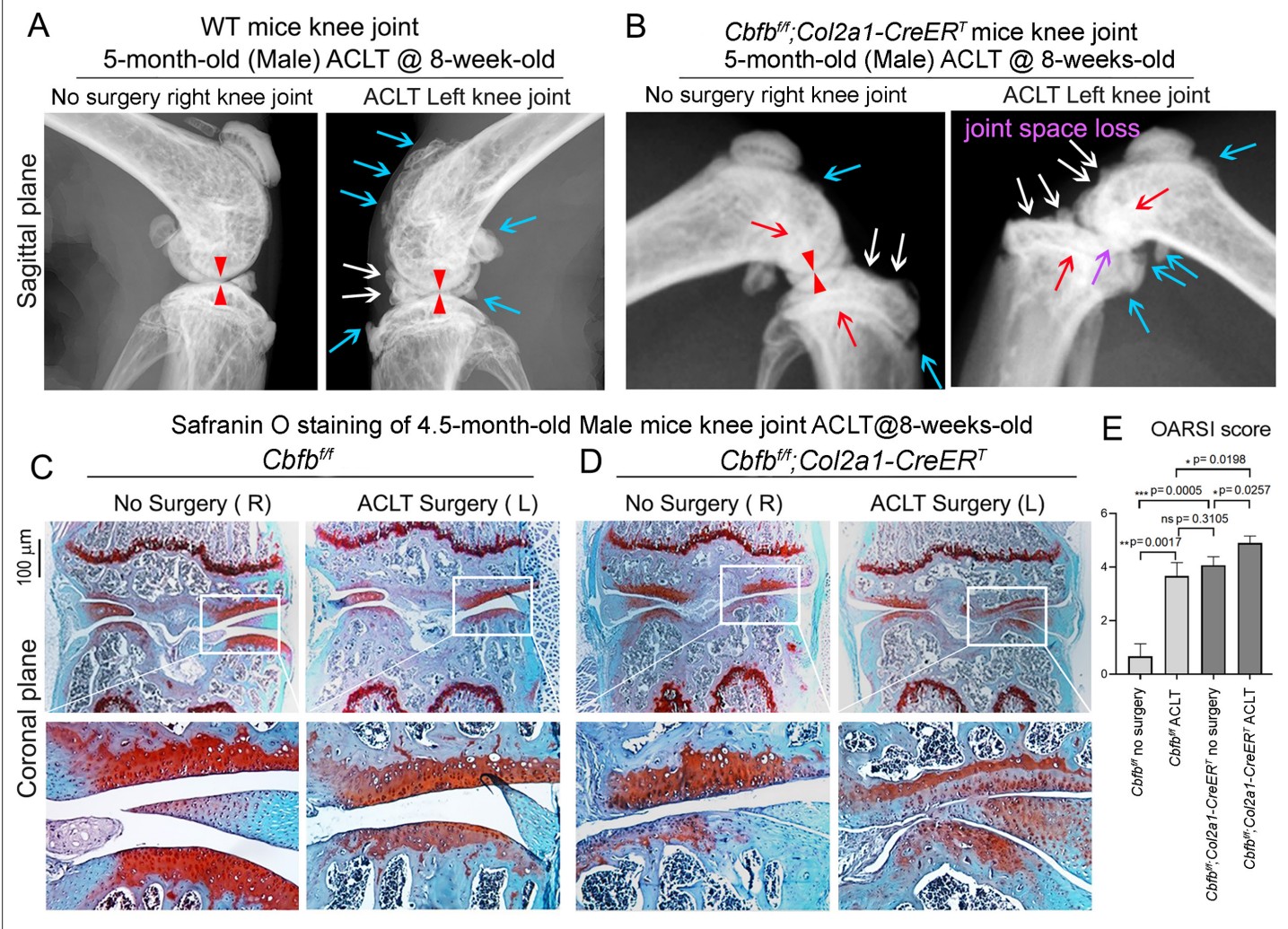

**Figure 3.** *Cbfb^f/f;Col2a1-CreER^T* mice with ACLT surgery developed early onset OA. (**A**) X-ray of 5-month-old male WT (ACLT at 8-week-old) mice knees (n=15). (**B**) X-ray of 5-month-old male *Cbfb^f/f;Col2a1-CreER^T* (ACLT at 8-week-old) mice knees. Red arrows indicate subchondral bone; Red arrow heads indicate joint space; Light blue arrows indicate osteophytes; White arrows indicate worn articular cartilage; Purple arrow indicates joint space loss; (n=15). (**C**) SO stain of 4.5-month-old male *Cbfb^f/f* (ACLT at 8-week-old) mice knees (n=7). (**D**) SO stain of 4.5-month-old male *Cbfb^f/f;Col2a1-CreER^T* (ACLT at 8-week-old) mice knees (n=6). (**E**) Knee joint Osteoarthritis Research Society International (OARSI) score of (**C**) and (**D**). Data are shown as mean ± SD. Scale bar: 100 μm (**C–D**).

arrow) compared to the control groups, indicating spontaneous OA-like symptoms (**Figure 3A and B**). Moreover, *Cbfb^f/f;Col2a1-CreER^T* mice with no ACLT surgery knee joint space has narrower joint space compared to WT mice with ACLT surgery (Red Arrowhead; **Figure 3A and B**). Those results show that Cbfβ deficiency accelerated the development of OA in the *Cbfb^f/f;Col2a1-CreER^T* mice with ACLT surgery. Moreover, SO staining also showed that *Cbfb^f/f;Col2a1-CreER^T* mice with ACLT surgery had less SO-positive area compared to *Cbfb^f/f* mice with ACLT surgery, indicating increased cartilage loss (**Figure 3C and D**). The Osteoarthritis Research Society International (OARSI) Score analysis showed that *Cbfb^f/f;Col2a1-CreER^T* TMX injected mice with no surgery presented similar OARSI Score compared with *Cbfb^f/f* mice with ACLT surgery, indicating the important role of Cbfβ in articular cartilage homeostasis (**Figure 3E**). Interestingly, *Cbfb^f/f;Col2a1-CreER^T* TMX mice with ACLT surgery had a significantly increased OARSI Score compared to *Cbfb^f/f;Col2a1-CreER^T* TMX injected mice with no surgery and *Cbfb^f/f* mice with ACLT surgery (**Figure 3E**). Those results indicate that Cbfβ also plays an important role in regulating postnatal cartilage regeneration as well as bone destruction in OA pathological condition, and demonstrated that the deficiency of Cbfβ *could* be the cause of early onset OA.

## RNA-seq analysis indicated that deficiency of Cbfβ in cartilage reduces cell fate commitment, cartilage regeneration and repair, and increases canonical Wnt signaling and inflammatory response

To dissect the mechanism underlying the role of Cbfβ in the articular cartilage regeneration in OA, genome-wide RNA-sequencing analysis was conducted using hip articular cartilage of 2-month-old *Cbfb^f/f;Col2a1-CreER^T* TMX injected mice compared with 2-month-old WT mice (*Figure 4*). Volcano plot results illustrated that the top downregulated genes included *Fabp3, Nmrk2, Csf3r, Rgs9, Plin5, Rn7sk*, and *Eif3j2*, whereas top upregulated genes included *Cyp2e1, Slc15a2, Alas2, Hba-a2, Lyve1, Snca, Serpina1b, Hbb-b1, Rsad2, Retn*, and *Trim10* in the articular cartilage of *Cbfb* conditional knockout mice (*Figure 4A*). Pie chart of articular cartilage from *Cbfb^f/f;Col2a1-CreER^T* mice demonstrated significantly altered differentially expressed genes (DEGs), where 70.7% were upregulated and 29.3% were downregulated (*Figure 4B*). Among them, *Rsad2* is known to be closely related to immune regulation and play a role in driving the inflammatory response through the NF-κB and JAK-STAT pathways (*Lin et al., 2013*). Increased expression of *Rsad2* indicates that *Cbfb* conditional knockout is associated with increased inflammatory signaling in mice knee joints (*Figure 4A*). Moreover, several genes related to lipid metabolism and transport were downregulated in response to *Cbfb* conditional knockout (*Figure 4A*). *Fabp3* is known to be involved in several processes, including lipid homeostasis and transport, and positive regulation of long-chain fatty acid import into cell (*Lee et al., 2020*; *Liu et al., 2021*). In addition, *Plin5* is a negative regulator of peroxisome proliferator activated receptor (PPAR) signaling, a positive regulatory of sequestering of triglyceride and regulation of lipid metabolic process (*Miner et al., 2023*). A previous study has shown that dysregulated lipid content or metabolism in cartilage leads to dysfunction cartilage (*Villalvilla et al., 2013*). Decreased expression of *Fabp3* and *Plin5* in *Cbfb* conditional knockout mice indicates the important positive regulatory role of Cbfβ in lipid transport and metabolism in articular cartilage, which is important in cartilage homeostasis (*Figure 4A*).

To further investigate the functions of the differential expressed genes in *Cbfb* conditional knockout mice, Gene Ontology (GO) studies were performed on both the upregulated and downregulated DEGs in *Cbfb^f/f;Col2a1-CreER^T* mice TMX injected compared to WT mice (*Figure 4C–F*). GO annotation based on GO Biological Processes (BP) showed significantly downregulated differentially expressed gene groups associated with Cellular Response to Retinoic Acid, Wound Healing, positive Regulation of Protein Phosphorylation in response to *Cbfb* conditional knockout in *Cbfb^f/f;Col2a1-CreER^T* mice, further supporting the important role of Cbfβ in cartilage and bone development (*Figure 4C*). Moreover, it was previously reported that p38/ERK/JNK/SMAD pathways are crucial in the chondrogenic differentiation induced by Tgfb1 (*Ma et al., 2019*). GO BP analysis also revealed significantly downregulated genes in positive regulation of ERK1 and ERK2 cascade in *Cbfb^f/f;Col2a1-CreER^T* mice, indicating that Cbfβ deficiency in chondrocytes was associated with downregulated ERK signaling which resulted in dysregulated chondrocyte differentiation (*Figure 4C*). Furthermore, *Cbfb* conditional knockout is also associated with downregulated cell fate commitment, cell differentiation, positive regulation of gene expression, animal organ morphogenesis, regulation of RNA polymerase II promoter, and positive regulation of protein phosphorylation (*Figure 4C*). Enrichment analysis of downregulated KEGG signaling pathways also demonstrated that Cbfβ deficiency in *Cbfb^f/f;Col2a1-CreER^T* mice led to significant changes in signaling pathways regulating pluripotency of stem cells (*Figure 4E*). These results implied that Cbfβ deficiency leads to downregulated chondrocyte differentiation and proliferation. In addition, downregulated differential expressed genes in Cbfβ deficient mice were associated with cellular response to insulin stimulus (*Figure 4C*). Previous studies have shown that insulin has anti-inflammatory effect by negatively regulating NF-κB, PI3k/AKT, and TLR signaling etc (*Tilich and Arora, 2011*; *Zhang et al., 2016*). Downregulated cellular response to insulin stimulus in *Cbfb^f/f;Col2a1-CreER^T* mice hip articular cartilage suggested dysregulated and elevated immune signaling in Cbfβ-deficient mice articular cartilage. In addition, a recent study has shown that the activated cAMP pathway inhibits OA development (*Xie et al., 2023*). cAMP signaling is downregulated in Cbfβ deficient mice, indicating that Cbfβ may also regulate cAMP signaling in OA pathogenesis (*Figure 4E*). On the other hand, top downregulated GO KEGG analysis in Cbfβ-deficient cartilage also shows downregulated PPAR signaling, in line with decreased Plin5 expression shown in the volcano plot (*Figure 4A and E*). As mentioned previously, PPAR signaling is crucial for cell differentiation and lipid metabolism (*Dunning et al., 2014*; *Feige et al., 2006*).

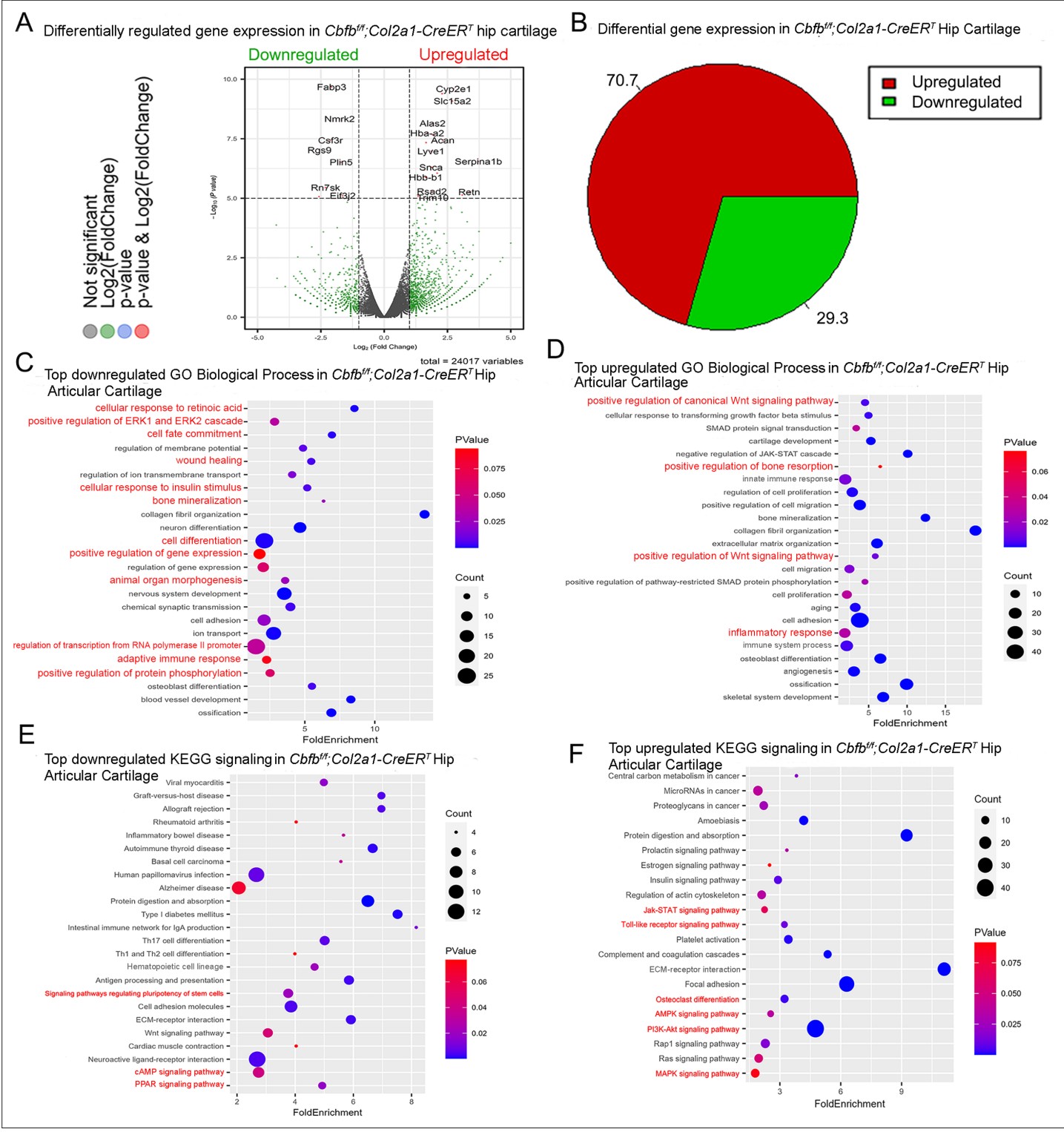

**Figure 4.** RNA-seq analysis indicated that deficiency of Cbfβ in cartilage reduces cell fate commitment, cartilage regeneration and repair, and increases canonical Wnt signaling and inflammatory response. (**A**) Volcano plot showing differentially regulated gene expression in 6-week-old male *Cbfb^{f/f}* and *Cbfb^{f/f};Col2a1-CreER^T* mice hip articular cartilage. (**B**) Pie chart showing percentage of upregulated and downregulated differentially regulated genes in hip articular cartilage of 6-weeks-old male *Cbfb^{f/f};Col2a1-CreER^T* mice compared to those of *Cbfb^{f/f}* mice. The percentages of genes upregulated and downregulated are shown in red and green, respectively. (**C**) GO functional clustering of the top downregulated biological process (BP) in 6-week-old male *Cbfb^{f/f};Col2a1-CreER^T* mice hip articular cartilage. (**D**) GO functional clustering of the top upregulated BP in 6-week-old male *Cbfb^{f/f};Col2a1-CreER^T* mice hip articular cartilage. (**E**) GO functional clustering of the top downregulated KEGG signaling pathways in 6-week-old male *Cbfb^{f/f};Col2a1-CreER^T*

*Figure 4 continued on next page*

Figure 4 continued

mice hip articular cartilage. (**F**) GO functional clustering of the top upregulated KEGG signaling pathways in 6-week-old male *Cbfb^{f/f};Col2a1-CreER^T* mice hip articular cartilage.

Downregulated PPAR signaling indicated a crucial role of Cbfβ in articular cartilage regeneration and regulation of lipid content.

Upregulated GO BP and GO KEGG analysis results further elucidated the regulatory mechanism of Cbfβ in mice articular cartilage (*Figure 4*). Firstly, upregulated GO BP pathways displayed significantly upregulated differentially expressed genes in the positive regulation of the canonical Wnt signaling pathway, indicating that Cbfβ negatively regulated canonical Wnt signaling pathway in articular cartilage (*Figure 4D*). Besides, upregulated positive regulation of bone resorption in *Cbfb* conditional knockout mice supported the bone destruction seen in previous phenotypical studies, showing the crucial role of Cbfβ in protecting against bone destruction (*Figure 4D*). Further, both GO BP and GO KEGG results unveiled upregulated signaling pathways related to inflammatory response (*Figure 4D and F*). Upregulated GO BP pathways including innate immune response, inflammatory response, immune system process, and angiogenesis were associated with Cbfβ deficiency and are also related to inflammation (*Figure 4D*). Furthermore, enrichment analysis of upregulated KEGG signaling pathways demonstrated that Cbfβ deficiency led to significant changes in the JAK-STAT signaling pathway, Toll-like receptor signaling pathway, AMPK signaling pathway, and MAPK signaling pathway (*Figure 4F*). The JAK/STAT pathway played an important role in multiple crucial cellular processes such as the induction of the expression of some key mediators that were related to cancer and inflammation (*Hu et al., 2021*). Moreover, studies had demonstrated the upregulation of TLR signaling in osteoarthritis (OA), highlighting its involvement in the induction of chondrocyte apoptosis (*Barreto et al., 2020*), along with the pivotal role played by MAPK signaling in the pathogenesis of OA (*Lan et al., 2021*). These GO data indicated an augmentation in the positive regulation of the JAK-STAT cascade, TLR signaling, and MAPK signaling pathways following *Cbfb* deletion, suggesting that the deficiency of Cbfβ led to an intensification of immune signaling contributing to the progression of osteoarthritic pathological processes. In addition, the downregulation of the adaptive immune response and the upregulation of the innate immune response further demonstrated that Cbfβ deficiency in the knee joint of mice was associated with heightened innate immune signaling while concurrently dampening adaptive immune signaling (*Figure 4C and D*).

## Heatmap analysis uncovered that Cbfβ deficiency in cartilage resulted in decreased chondrocyte genes expression and decreased Tgfβ and Hippo signaling, but increased Wnt signaling

To further uncover the regulatory mechanism by which Cbfβ initiates signaling pathway changes in OA at the individual gene level, the gene expression profiles associated with chondrocytes, as well as with the Hippo, Tgfβ, and Wnt signaling pathways were examined (*Figure 5*). Given that OA is a systemic joint disease, an analysis was conducted on both the articular cartilage of the hip joint in *Cbfb^{f/f};Col2a1-CreER^T* mice and the articular cartilage of the knee joint in *Cbfb^{f/f};Acan-CreER^T* mice (*Figure 5*). Interestingly, chondrocyte-related genes were downregulated in the hip joint articular cartilage of the *Cbfb^{f/f};Col2a1-CreER^T* mice, while upregulated genes in the *Cbfβ*-deficient mice knee articular cartilage included *Bmp2* and *Runx2* (*Figure 5A*). The Cbfβ subunit is a non-DNA-binding protein that binds Cbfα (also known as Runx) proteins to mediate their DNA-binding affinities. Runx/Cbfβ heterodimers play key roles in various developmental processes (*Wu et al., 2014a*; *Wu et al., 2014b*; *Westendorf and Hiebert, 1999*; *Tian et al., 2014*; *Chen et al., 2014*; *Park et al., 2016*; *Qin et al., 2015*; *Lim et al., 2015*). Moreover, Bmp2 is a crucial protein in the development of bone and cartilage, a central protein in Tgfβ signaling, and some of its specific functions include activating osteogenic genes such as *Runx2* (*Halloran et al., 2020*). Many genes were also upregulated in the Cbfβ-deficient articular cartilage, such as *Adamts12* and *Fgf2* (*Figure 5A*). High expression of Adamts is a typical feature of OA, implying that Cbfβ deficiency may control the expression of Adamts to affect the differentiation of chondrocytes. Further, Fgf2 is previously reported to activate Runx2 via MER/ERK signaling pathway and increase MMP13 expression (*Wang et al., 2004*). Increased expression of

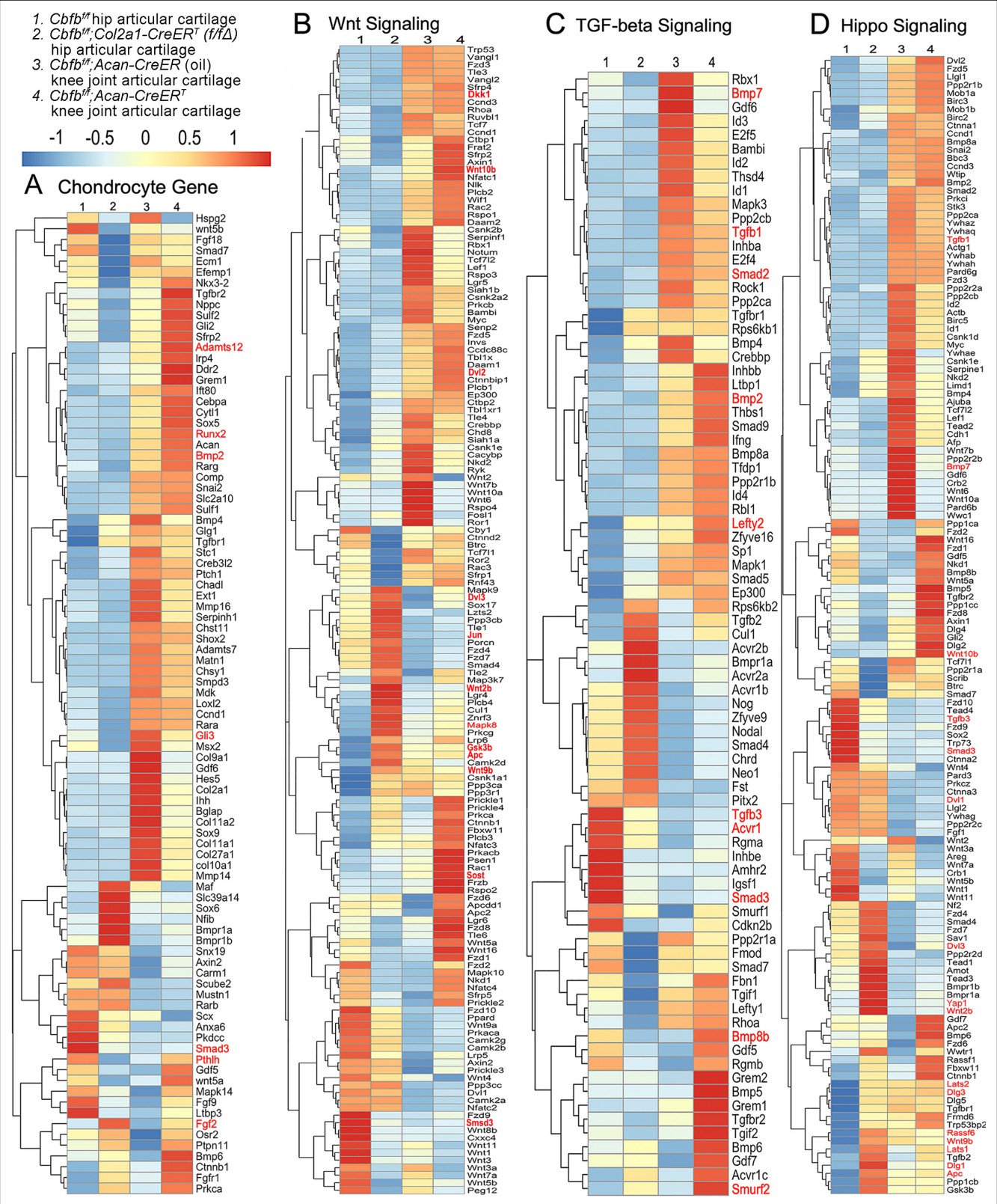

**Figure 5.** Heatmap analysis uncovered that deficiency of Cbfβ in cartilage resulted in decreased chondrocyte genes expression and decreased Tgfβ and Hippo signaling, but increased Wnt signaling. (**A**) Heatmap for chondrocyte gene expression in (1) 6-weeks-old male *Cbfb^f/f* mice hip articular cartilage, (2) 6-week-old male *Cbfb^f/f;Col2a1-CreER^T* mice hip articular cartilage, (3) 12-week-old male oil injected *Cbfb^f/f;Acan-CreER^T* mice knee joint articular cartilage, and (4) 12-week-old male *Cbfb^f/f;Acan-CreER^T* mice (TMX injected at 6-week-old) knee joint articular cartilage. (**B**) Heatmap showing Wnt

*Figure 5 continued on next page*

*Figure 5 continued*
signaling-related gene expression. (**C**) Heatmap showing Tgfβ signaling-related gene expression. (**D**) Heatmap showing Hippo signaling-related gene expression.

*Fgf2* was seen in both *Cbfb^{f/f};Col2a1-CreER^T* hip articular cartilage as well as *Cbfb^{f/f};Acan-CreER^T* knee cartilage, showing Cbfβ might upregulate MAPK/ERK signaling in cartilage through Fgf2 (*Figure 5A*).

Moreover, the heatmap of RNA-seq analysis showed that *Cbfb^{f/f};Col2a1-CreER^T* cartilage had altered gene expression levels in the Wnt, Tgfβ, and Hippo signaling pathways (*Figure 5*). Our results demonstrated that genes associated with Wnt signaling pathway activation, such as *Mapk8*, *Dvl3*, *Wnt10b*, *Wnt2b*, *Wnt9b*, and *Jun* (*Nie et al., 2020*) were upregulated, whereas the inhibitor of the Wnt signaling pathway *Sost* was downregulated, indicating that loss of *Cbfb* could promote cartilage ossification and osteophyte formation through its activation of the Wnt pathway (*Figure 5B*). Dvl3 is a positive regulator of the Wnt/b-catenin pathway, which can stabilize β-catenin and upregulate downstream target genes by interacting with Mex3a (*Yang et al., 2022*). These results suggested that the loss of *Cbfb* could promote the expression of the activator of Wnt signaling, resulting in the activation of the Wnt signaling pathway.

Furthermore, our results also exemplified that the Tgfβ signaling pathway repressors *Lefty2* and *Smurf2* were upregulated in the Cbfβ-deficient articular cartilage (*Chandhoke et al., 2016*; *Figure 5C*). In addition, other genes involved in Tgfβ signaling, such as *Tgfb1*, *Acvrl1*, *Bmp7*, *Smad2*, and *Smad3*, were downregulated in the Cbfβ-deficient articular cartilage (*Figure 5C*). These results demonstrate that loss of *Cbfb* leads to decreased expression of genes in Tgfβ signaling and increased expression of repressors of Tgfβ signaling, which results in the inhibition of the Tgfβ signaling pathway.

Genes involved in the canonical Hippo signaling pathway such as *Apc*, *Dlg1*, and *Dlg3* were upregulated, signifying a close relationship of *Cbfb* to Hippo signaling (*Figure 5D*). Apc is the downstream part of the Wnt signaling pathway, and through the cross-talk of Wnt signal and Hippo signal, Apc mutation leads to the nuclear localization of Yap/Taz and activates Yap-Tead and Taz-Tead-dependent transcription, and ultimately, Hippo signal is turned off (*Azzolin et al., 2014*). In our study, *Apc* expression was enhanced in Cbfβ-deficient articular cartilage, supporting that Cbfβ deficiency in articular cartilage affected Hippo signaling (*Figure 5D*). Lats1 and Lats2 are essential components of the Hippo pathway that phosphorylate and inactivate Yap, which is a key link in the activation and shutdown of the Hippo signaling pathway (*He et al., 2019*). Our study demonstrated that *Lats1/Lats2* expression was enhanced in Cbfβ-deficient articular cartilage (*Figure 5D*). Therefore, although there is increased *Yap1* gene expression in Cbfβ-deficient mice, upregulated *Lats1/Lats2* potentially leads to increased phosphorylation in Yap protein and activated Hippo signaling pathway (*Figure 5D*). Thus, loss of *Cbfb* could inhibit the repressor of the Hippo signaling pathway and promote the expression of the activator of Hippo signaling, resulting in the activation of the Hippo signaling pathway. Examination of the expression profiles of these genes showed altered expression between the mutant and WT samples, with different expression patterns between Cbfβ-deficient articular cartilage in mice hip samples and Cbfβ-deficient knee samples, indicating that Cbfβ regulation is tissue-specific (*Figure 5A–D*). Collectively, we are the first to demonstrate that Cbfβ may control downstream gene expression by orchestrating the Tgfβ, Hippo, and Wnt signaling pathways, thereby setting off the cascade of OA pathological processes, including cartilage damage and inflammation.

## Postnatal Cbfβ deficiency in cartilage resulted in increased Wnt signaling, inflammatory genes expression, decreased cartilage formation genes expression in the knee articulate cartilage

To further investigate OA-related genes expression of *Cbfb* deficiency mice in articular cartilage in which Cbfβ regulates articular cartilage regeneration, we performed immunohistochemistry (IHC) staining on Cbfβ-deficient mouse hip joints. The result showed that postnatal Cbfβ deficiency in cartilage (*Figure 6A and F*) resulted in increased inflammatory markers expression and decreased cartilage formation markers expression in the knee articulate cartilage. The chondrocytes cell markers Col2α1, and cartilage degradation markers MMP13 and ADAMTS5 were examined by IHC staining (*Figure 6B, C, D and G*). As expected, mutant mice articular cartilage had significant degradation with low expression of Col2α1 in both the superficial zone and the deep zone, and the middle zone was

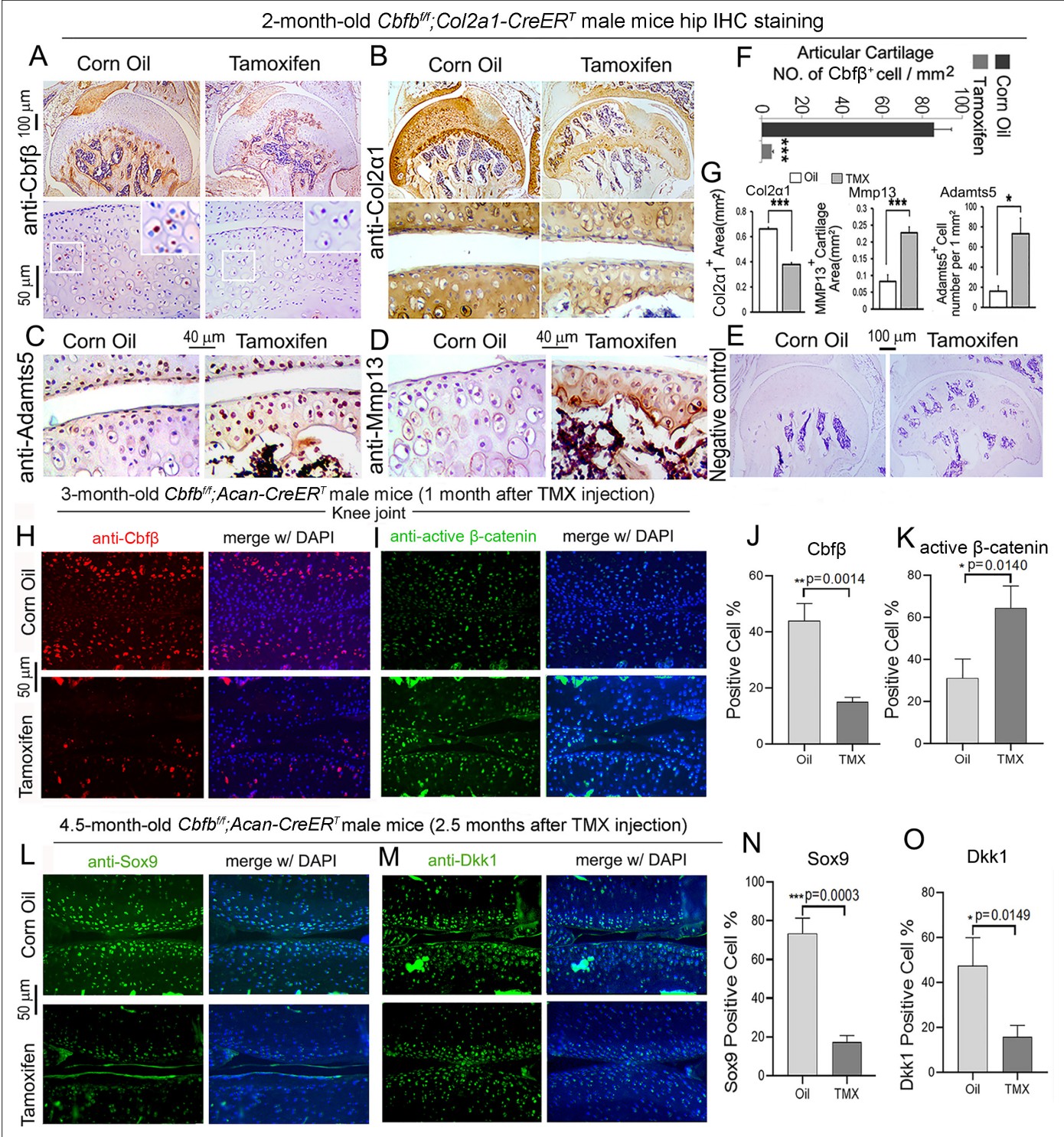

**Figure 6.** Postnatal Cbfβ deficiency in cartilage resulted in increased Wnt signaling, inflammatory genes expression, decreased cartilage formation genes expression in the knee articulate cartilage. (**A–E**) IHC staining of (**A**) anti-Cbfβ, (**B**) anti-Col2α1, (**C**) anti-Adamts5, and (**D**) anti-Mmp13 of hip joint from 2-month-old male *Cbfb^{f/f};Col2a1-CreER^T* mice. (**E**) Negative control of (**A–D**). (**F**) Quantification for (**A**). (**G**) Quantification for (**B–D**). (**H–I**) IF staining of (**H**) anti-Cbfβ and (**I**) Active-β-catenin of knee joint from 3-month-old male *Cbfb^{f/f};Acan-CreER^T* mice. (**J–K**) Quantification of (**H**) and (**I**). (**L–M**) IF staining of (**L**) anti-Sox9, and (**M**) anti-Dkk1 of knee joint from 4.5-month-old male *Cbfb^{f/f};Acan-CreER^T* mice with oil injection or TMX injection. (**N–O**) Quantification of (**L**) and (**M**). Data are shown as mean ± SD. n=3. *p<0.05, **p<0.01, ***p<0.001.

replaced by bone with no Col2α1 expression (*Figure 6B and G*). Aggrecanases (Adamts) and matrix metalloproteinases (MMPs), especially Adamts5 and Mmp13 are known to have important roles in cartilage destruction in OA. IHC staining results show without Cbfβ, articular cartilage had high levels of Adamts5 (*Figure 6C and G*) and Mmp13 (*Figure 6D and G*), indicating mutant mice cartilage was undergoing severe cartilage degradation and increased inflammation. Negative control of the IHC staining shows the validity of the experiment (*Figure 6E*).

As previous data had shown that Cbfβ-deficiency impaired articular cartilage regeneration, signaling pathways that regulate OA was our next focus, such as Wnt signaling. IF staining showed efficient *Cbfb* deletion in mouse articular cartilage (*Figure 6H and J*). Moreover, IF staining of active β-catenin showed that in knee joint articular cartilage of *Cbfb^{f/f};Acan-CreER^T* mice, there is increased Active-β-catenin expression compared to control (*Figure 6I and K*). This confirmed that Cbfβ has an important role in regulating Wnt/β-catenin signaling pathway. IF staining of 4.5-month-old oil-injected *Cbfb^{f/f};Acan-CreER^T* and TMX-injected *Cbfb^{f/f};Acan-CreER^T* mice knee joints articular cartilage further exhibited that Sox9 protein level was decreased in the Cbfβ-deficient joint (*Figure 6L and N*). As Sox9 is involved in articular cartilage formation, this observation suggests that Cbfβ is involved in regulating articular cartilage formation (*Figure 6L and N*). Dickkopf-1 (Dkk1), a Wnt signaling inhibitor, was decreased expression in knee joints articular cartilage of 4.5-month-old *Cbfb^{f/f};Acan-CreER^T* mice, indicating that Cbfβ plays a role in regulating articular cartilage homeostasis through the Wnt signaling pathway by inhibiting Dkk1 (*Figure 6M and O*).

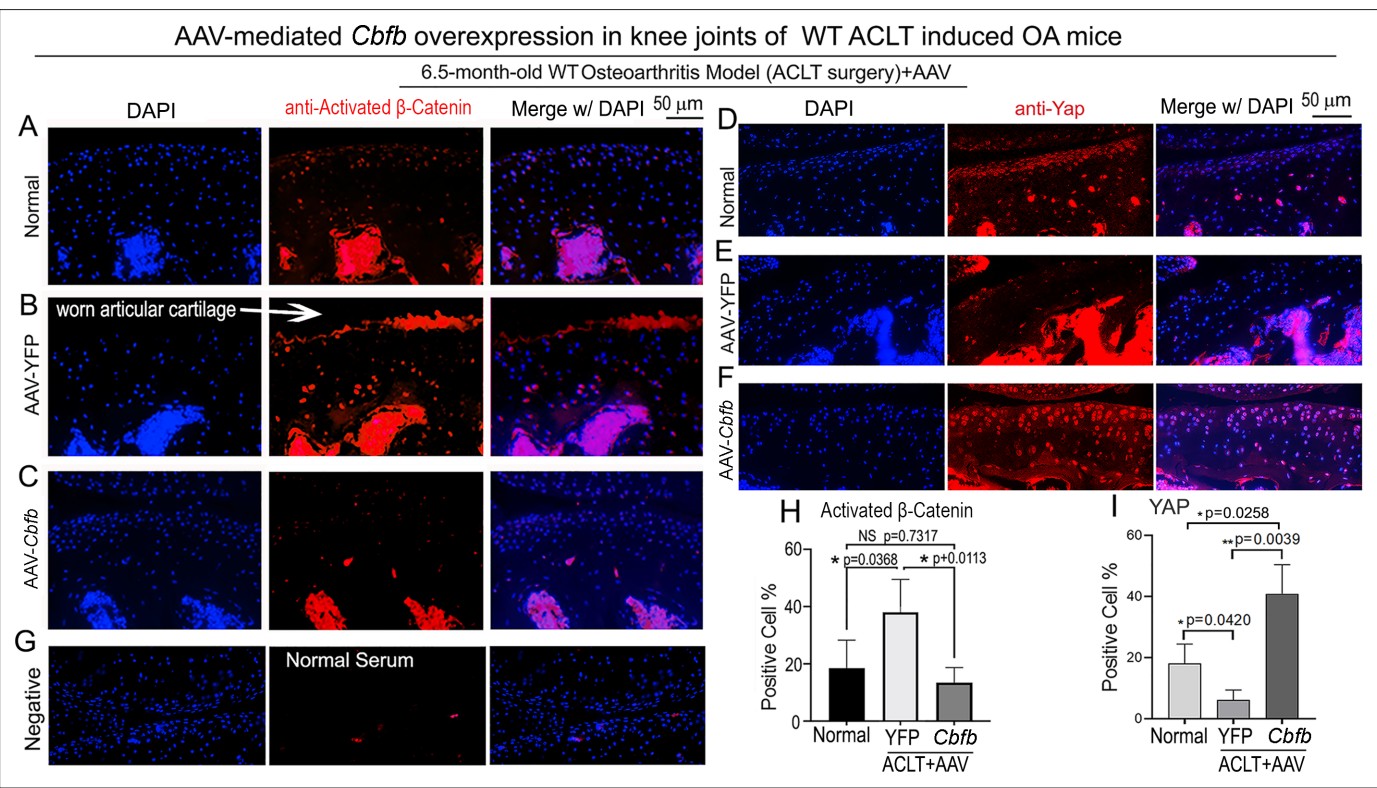

**Figure 7.** Locally administrated AAV-mediated *Cbfb* overexpression inhibited β-Catenin expression and enhanced Yap expression in knee joints articular cartilage of ACLT-induced OA mice. (**A–C**) IF staining of anti-active-β-catenin in the knee joints articular cartilage of 6.5-month-old male (**A**) Normal WT, (**B**) AAV-YFP with ACLT surgery, and (**C**) AAV-*Cbfb* mice with ACLT surgery (n=3). (**D–F**) IF staining of anti-Yap in the knee joints articular cartilage of 6.5-month-old (**D**), (**E**) AAV-YFP ACLT surgery, and (**F**) AAV-*Cbfb* mice with ACLT surgery (n=3). (**G**) Negative control of (**A–F**). (**H**) Quantification of (**A–C**). (**I**) Quantification of (**D–F**). Data are shown as mean ± SD.

The online version of this article includes the following figure supplement(s) for figure 7:

**Figure supplement 1.** Successful AAV-luc-YFP infection in mice.

## Locally administrated AAV-mediated *Cbfb* overexpression inhibited β-Catenin expression and enhanced Yap expression in knee joints articular cartilage of ACLT-induced OA mice

To further characterize the mechanism underlying Cbfβ regulates articular cartilage in both physiological conditions and pathological conditions, we applied locally administrated AAV-mediated *Cbfb* overexpression as a Gain-of-Function approach. We first proved locally administrated AAV can successfully infiltrate knee joints articular cartilage by using AAV-luc-YFP infection in mice (*Figure 7—figure supplement 1*). We then analyzed β-Catenin expression and Yap expression at the knee joints articular cartilage of 6.5-month-old WT mice with ACLT surgery that were either administered AAV-YFP as control or AAV-*Cbfb* by intra-articular injection (*Figure 7*). AAV-mediated *Cbfb* overexpression decreased about 2.5-fold Active-β-catenin expression at the knee joints articular cartilage compared to AAV-YFP ACLT group (*Figure 7A, B, C and H*). AAV-mediated *Cbfb* overexpression increased Yap expression about 3.5-fold in the ACLT knee joints articular cartilage compared to the AAV-YFP ACLT group (*Figure 7D, E, F, I*). These results from the Gain-of-Function approach confirmed that Cbfβ regulates Wnt/β-catenin and Hippo/Yap signaling pathways in articular cartilage homeostasis and suggests that local over-expression of *Cbfb* could be an effective target for OA treatment.

## deficiency of Cbfβ decreased the expression of Yap, and Smad2/3 and increased Mmp13 expression, and overexpression of *Cbfb* increased Yap expression and decreased β-catenin expression

To further explore the regulatory mechanism through in vitro studies, we used Alcian Blue staining of primary chondrocytes prepared from newborn *Cbfb^{f/f};Col2a1-Cre* mice growth plates and showed significantly reduced matrix deposition in mutant chondrocytes, which was reflected by weaker Alcian Blue staining of the cells on days 7, 14, and 21 (*Figure 8—figure supplement 1*). Moreover, *Cbfb* overexpression in ATDC5 (chondrocyte cell line) showed about twofold increased Yap protein level compared to control (*Figure 8A and B*).

We next examined the Cbfβ, p-Smad2/3, Smad2/3 and Mmp13 protein level changes in hip articular cartilage in TMX injected *Cbfb^{f/f};Col2a1-CreER^T* mice. The western blot of the hip cartilage samples showed about 3.5-fold, 10-fold and 3.5-fold decrease in protein levels of Cbfβ, p-Smad2/3, and Smad2/3 respectively, and a 10-fold increase in the protein level of Mmp13 (*Figure 8C and F*). To determine whether ACLT induced OA affects Cbfβ protein levels, we detected Cbfβ protein in the knee joint articular cartilage of ACLT-induced OA of 16-week-old male WT mice. The result showed that Cbfβ protein in WT mice with ACLT surgery decreased by about twofold compared to the no ACLT WT mice control, and Cbfβ protein was decreased by about fourfold in *Cbfb^{f/f};Col2a1-CreER^T* mice with ACLT compared to the control (*Figure 8D and G*). This result indicated that low expression of Cbfβ may be a cause of OA pathogenesis.

To further characterize the mechanism by which Cbfβ regulates β-catenin expression in ACLT induced OA, the protein samples from the knee joint articular cartilage of 16-week-old male WT mice with ACLT treated with AAV-*Cbfb* overexpression were analyzed by western blot which showed about twofold increased Cbfβ protein level in the knee joint articular cartilage of 16-week-old male WT mice with ACLT treated with AAV-*Cbfb*-mediated overexpression (*Figure 8E and H*), and about 10-fold decreased protein level in Active-β-catenin when compared to mice with no AAV-*Cbfb* treatment control (AAV-YFP; *Figure 8E and H*). Together, these data show that Cbfβ plays a central role in regulating the hip and knee joints articular cartilage homeostasis through Wnt/β-catenin, Hippo/Yap and Tgfβ signaling pathways.

## Adeno-associated virus (AAV)-mediated *Cbfb* overexpression protects against ACLT-induced OA

To investigate the therapeutic effect of Cbfβ in ACLT induced OA, AAV-*Cbfb* was locally administrated for AAV-mediated *Cbfb* overexpression in knee joints articular cartilage of ACLT-induced OA mice. We performed X-rays and SO staining on WT mice with or without ACLT surgery and with either no treatment, AAV-YFP control treatment, or AAV-*Cbfb* treatment (*Figure 9*). In the X-ray images, yellow arrows indicate normal joint space; white arrows indicate worn articular cartilage; blue arrows indicate osteophytes; red arrows indicate joint space loss (*Figure 9A, B*). We observed that 22-week-old male

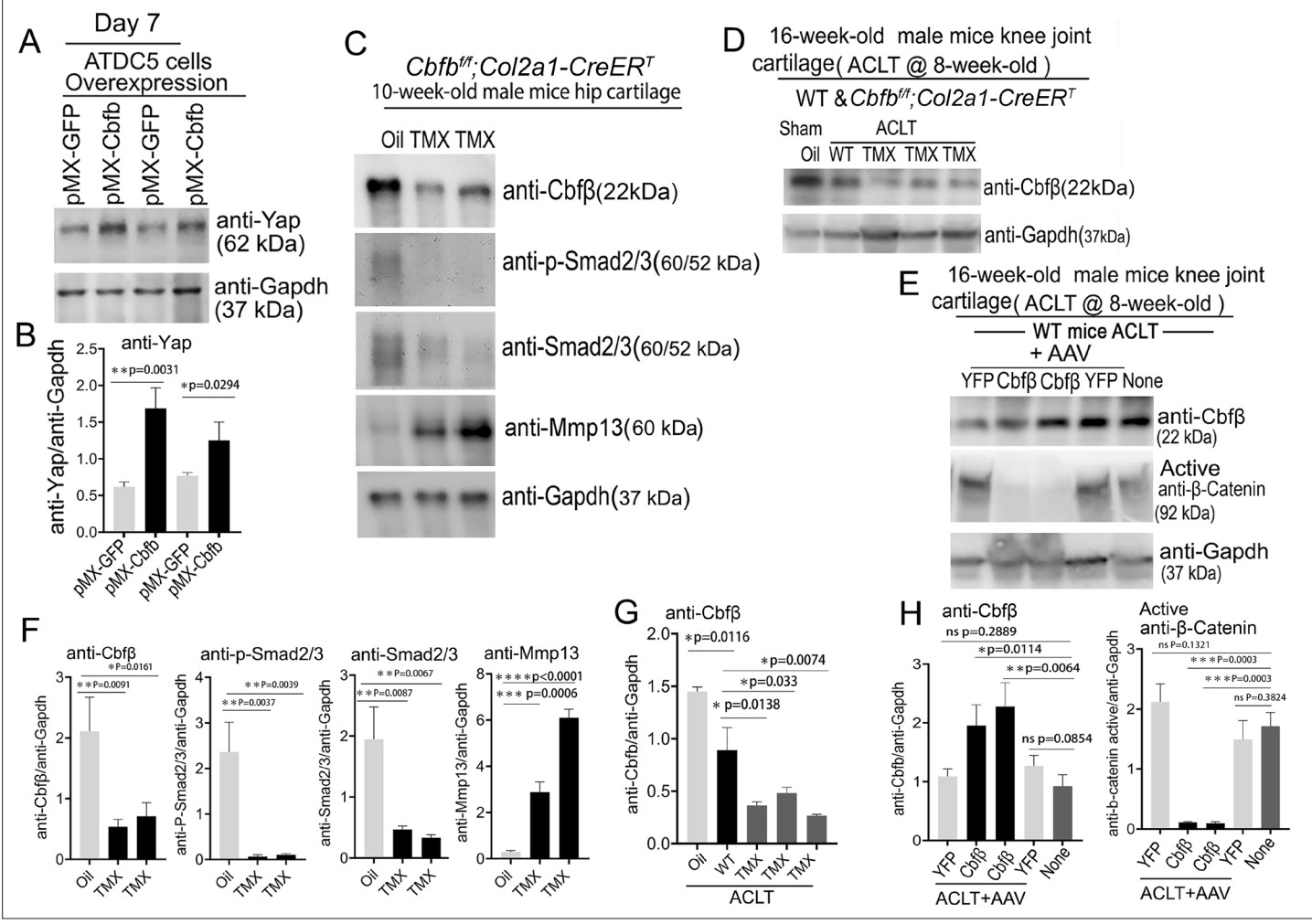

**Figure 8.** Deficiency of Cbfβ protein levels increased β-catenin and articular cartilage degradation markers while also reducing Yap signaling activation. (**A**) Western blot showing protein expression level of Yap in ATDC5 cells (n=3). (**B**) Quantification of Yap protein levels in (**A**). (**C**) Western blot of 10-week-old male hip cartilage from *Cbfb^{f/f};Col2a1-CreER^T* mice injected with either oil or TMX showing the expression of Cbfβ, p-Smad2/3, Smad2/3, and Mmp13 (n=5). (**D**) Western blot of knee joint cartilage from 16-week-old male WT and *Cbfb^{f/f};Col2a1-CreER^T* mice with ACLT surgery and injected with either oil or TMX showing the expression of Cbfβ (n=6). (**E**) Western blot of WT mice knee joint cartilage from 16-week-old male mice with ACLT surgery, treated with AAV-luc-YFP or AAV-*Cbfb*, and injected with either oil or TMX showing the expression of Cbfβ and active β-catenin (n=6). (**F**) Quantification of (**C**). (**G**) Quantification of (**D**). (**H**) Quantification of (**E**). Data are shown as mean ± SD. *p<0.05, **p<0.01, ***p<0.001. NS Not Significant.

The online version of this article includes the following source data and figure supplement(s) for figure 8:

**Source data 1.** Labeled raw western blot data for *Figure 8A* (anti-Yap and anti-Gapdh).

**Source data 2.** Unlabeled raw western blot data for *Figure 8A* (anti-Yap and anti-Gapdh).

**Source data 3.** Labeled raw western blot data for *Figure 8C* (anti-Cbfβ, anti-p-Smad2/3, anti-Smad2/3, anti-Mmp13, and anti-Gapdh).

**Source data 4.** Unlabeled raw western blot data for *Figure 8C* (anti-Cbfβ, anti-p-Smad2/3, anti-Smad2/3, anti-Mmp13, and anti-Gapdh).

**Source data 5.** Labeled raw western blot data for *Figure 8D* (anti-Cbfβ and anti-Gapdh).

**Source data 6.** Unlabeled raw western blot data for *Figure 8D* (anti-Cbfβ and anti-Gapdh).

**Source data 7.** Labeled raw western blot data for *Figure 8E* (anti-Cbfβ, anti-active β-catenin, and anti-Gapdh).

**Source data 8.** Unlabeled raw western blot data for *Figure 8E* (anti-Cbfβ, anti-active β-catenin, and anti-Gapdh).

**Figure supplement 1.** Alcian Blue staining of primary chondrocytes from Cbfβ deficient newborn mice show reduced matrix deposition.

WT mice with ACLT surgery developed an OA phenotype including unclear borders, narrow joint space, and hyperosteogeny (**Figure 9A, C**). We noticed that in the mice with ACLT surgery, the knee which was not operated on also developed a slight OA phenotype with narrow joint space. Notably, we observed that 22-week-old male WT mice with ACLT surgery treated with AAV-*Cbfb* did not develop

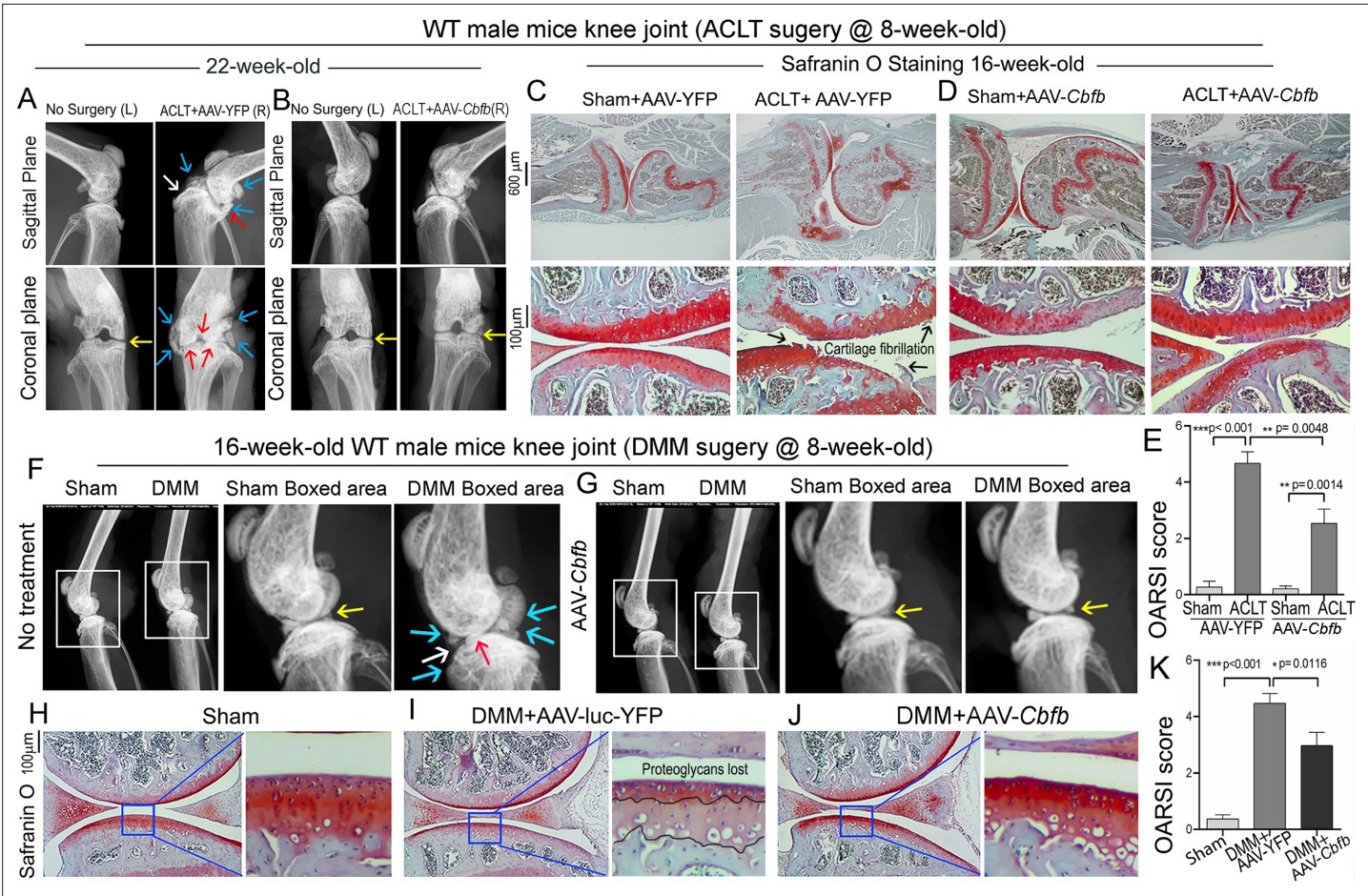

**Figure 9.** Adeno-associated virus (AAV)-mediated *Cbfb* overexpression protects against ACLT mechanical OA. (**A–B**) X-ray images of the knee joints of 22-week-old male WT mice with ACLT surgery at 8-week-old with (**A**) AAV-YFP treatment and (**B**) AAV-*Cbfb* treatment (n=15). Yellow arrows indicates normal joint space; White arrows indicate worn articular cartilage; blue arrows indicate osteophytes; red arrows indicate joint space loss. (**C–D**) SO staining of knees from 16-week-old male WT mice with (**C**) AAV-YFP (control) or (**D**) AAV-Cbfb treatment in ACLT mediated OA (ACLT surgery at 8-week-old) (n=5). (**E**) Knee joint of OARSI score of (**C**) and (**D**). (**F–G**) X-ray images of mouse knee joints of 16-week-old male mice after sham/DMM surgery with (**F**) no treatment or (**G**) AAV-*Cbfb* treatment (n=15). White arrows: osteophytes and worn articular cartilage. (**H–J**) SO staining of knee joints of 16-week-old mice after sham/DMM surgery (DMM surgery at 8-week-old) with (**H**) Sham no treatment, (**I**) DMM surgery AAV-YFP treatment, or (**J**) AAV-*Cbfb* treatment (n=5). (**K**) Knee joint OARSI score of (**H–J**). The results are presented as the mean ± SD, *p<0.05, **p<0.01, ***p<0.001. DMM surgery AAV-YFP treatment group shows severe cartilage damage, osteophytes, and delocalized knee joint, while the AAV-*Cbfb* treated group shows less cartilage loss and osteophytes than control.

unclear borders, narrow joint space, hyperosteogeny, or worn articular cartilage when compared to 22-week-old male WT mice with ACLT surgery that were treated with AAV-YFP (*Figure 9A and B*). To further investigate the role of Cbfβ in pathological OA through gain-of-function, we performed SO staining and histological analysis. AAV-mediated *Cbfb* overexpression treatments were administrated to ACLT surgery-induced OA mouse models. First, we performed ACLT surgery on 8-week-old WT mice administered with AAV-YFP as control or AAV-*Cbfb* by intra-articular injection. SO staining of the mice at 16-weeks-old showed severe articular cartilage loss in AAV-YFP-treated OA mice knees, with articular cartilage degradation and osteophytes, while the AAV-*Cbfb* treatment group had attenuated articular cartilage damage and significantly reduced OARSI scores compared to AAV-YFP control (*Figure 9C, D and E*). These data suggest that AAV-*Cbfb*-treated mice were protected from ACLT-induced OA damage compared to the control, and that local overexpression of *Cbfb* could be an effective therapeutic strategy for OA treatment.

We also used the surgical destabilization of the medial meniscus (DMM) surgery-induced OA model to test Cbfβ's role in protecting against OA. We observed that in DMM surgery-induced OA, the OA phenotype was evident (*Figure 9F and G*) as indicated by blue and white arrows. Notably,

in DMM surgery-induced OA with AAV-*Cbfb* treatment the OA phenotype was not seen, and only slightly increased subchondral bone density was observed (*Figure 9F and G*). To further investigate the role of Cbfβ in pathological OA through a gain-of-function approach, AAV-mediated *Cbfb* over-expression treatments were administrated to DMM surgery-induced OA mouse models. SO staining of the mice at 16-week-old showed articular cartilage loss in AAV-YFP treated OA mice knees, with degraded articular cartilage and osteophytes, while the AAV-*Cbfb* treatment group displayed atten-uated articular cartilage damage and significantly reduced OARSI scores compared to AAV-YFP control (*Figure 9G–K*). Consistent with our SO staining of ACLT model of OA, evaluation of mice knee joints in DMM-induced OA showed loss of articular cartilage, decreased joint space, and increased OARSI score (*Figure 9H, I, K*). Compared to AAV-YFP controls, treatment with AAV-*Cbfb* attenu-ated articular cartilage damage and significantly reduced OARSI scores compared to AAV-YFP control (*Figure 9I–K*). Thus, local overexpression of *Cbfb* could be a novel and effective target for the treat-ment of osteoarthritis.

## Discussion

In our current study, we showed that the deletion of *Cbfb* in postnatal mice cartilage caused severe spontaneous OA through Cbfβ's regulation of multiple key signaling pathways. We demonstrated that the changes of OA-related gene expression in the articular cartilage of aging-associated and *Cbfb* deficiency induced OA included downregulated Sox9, Dkk1, Yap, and p-Smad2/3, and upregu-lated Wnt5a and Wnt/β-catenin. We conclude that Cbfβ promotes articular cartilage regeneration and repair by modulating multiple key signaling pathways, including the Wnt/β-catenin, Tgfβ, and Hippo/Yap pathways.

Our previous studies have proven Cbfβ's important role in bone skeletal development (*Wu et al., 2014b*; *Chen et al., 2014*). Cbfβ and Runx1 plays crucial roles in regulating both chondrocytes and osteoblast in bone (*Wu et al., 2014a*; *Tang et al., 2020b*; *Zhang et al., 2022*; *Tian et al., 2014*; *Wu et al., 2017b*). Cbfβ is known to bind to Runx proteins (Runx1, Runx2, Runx3) through the Runt domain, and exon 5 of the *Cbfb* gene is essential for Cbfβ-Runx binding ability (*Park et al., 2016*). Recent studies revealed that Cbfβ played an important role in stabilizing Runx proteins (*Park et al., 2016*; *Qin et al., 2015*). Several OA susceptibility genes were identified through a genome-wide DNA methylation study in OA cartilage tissue, including Runx1 and Runx2 (*Jeffries et al., 2014*). Runx1 was reported to be highly expressed in knee superficial zone chondrocytes, to regulate cell proliferation (*LeBlanc et al., 2015*). In addition, Runx1 expression was increased in knees with OA (*LeBlanc et al., 2015*). Furthermore, *Runx1* mRNA injection showed a protective effect on surgically induced OA knees (*Aini et al., 2016*). We evaluated previously published datasets from human patients with OA (*Fisch et al., 2018*; *Rushton et al., 2014*) and found that *CBFB* expression is significantly decreased in OA, while *CBFB* methylation is significantly increased. These epigenetic changes could be the result of internal factors such as genetics or external factors such as diet, exercise, and environmental exposures. Interestingly, Li et al reported the relationship between abnormal *CBFB* expression in human cartilage and OA and reported that *CBFB* was highly expressed in the osteoarthritic cartilage (*Li et al., 2021*). However, in our study and Che et al's report (*Che et al., 2023*), *CBFB* expression in human OA was decreased compared to human normal cartilage. In our study, we found Runx1 expres-sion in the superficial zone and columnar chondrocytes of hip cartilage. In *Cbfb* deletion mice, *Runx1* expression was largely decreased at both the mRNA and protein level, which could directly relate to OA development. In contrast, Runx2 is known to promote OA formation by upregulating Mmp13 (*Wang et al., 2004*; *Kamekura et al., 2006*). However, a recent publication has shown that overex-pression of *Runx2* driven by the ColX promoter in mice showed delayed chondrocyte maturation and decreased susceptibility to develop OA, indicating that temporally and spatially different expressions of Runx2 may play opposite roles in OA (*Lu et al., 2014*).

While the Acan-creER[T] system, as employed in both current and previous studies (*Che et al., 2023*; *Zhang et al., 2024*), provides valuable insights into the role of Cbfβ in differentiated cartilage cells and its implications in the advanced stages of osteoarthritis, our current study also used *Cbfb*[f/f];*Col2a1-CreER*[T] aimed to explore the gene's function from a broader perspective. Previous study points out that Col2α1 is expressed in both early and late stage of chondrogenesis, including skeletal mesenchymal cells, perichondrium and presumptive joint cells, but aggrecan is expressed specifically in differentiated chondrocytes (*Blaney Davidson et al., 2014*). However, studies show that not only

differentiated chondrocytes but also chondrocyte progenitors are involved in OA pathogenesis (*Tong et al., 2022*). In our current study, the *Col2a1-CreER*^T system allowed us to investigate Cbfβ's role not only in mature chondrocytes but also in early chondroprogenitor cells, offering a comprehensive view of Cbfβ's involvement in cartilage in osteoarthritis. Therefore, the use of the *Cbfb*^f/f*;Col2a1-CreER*^T mouse mutant strain was instrumental in expanding our understanding of Cbfβ's multifaceted role in osteoarthritis, highlighting its importance not only in mature cartilage but also in the early stages of cartilage formation and differentiation. In addition to the different types of Cre used compared to our previous study, our current study also used gain-of-function approach in ACLT-induced OA disease model to understand the potential therapeutic function of Cbfβ in OA pathological condition. Adding our current findings to our previous research, we can now piece together a more complete picture of Cbfβ's role across the entire spectrum of cartilage development in osteoarthritis.

Our data indicated that *Cbfb* deletion upregulated the Wnt canonical signaling pathway during chondrocyte homeostasis, while local AAV-mediated *Cbfb* overexpression inhibited β-catenin expression and enhanced Yap expression in knee joints articular cartilage of ACLT-induced OA mice. RNA-seq analysis showed increased expression of Wnt10b, Wnt2b, the activator of Wnt signaling, and decreased Wnt antagonistic inhibitor Sost expression in *Cbfb*^f/f*;Col2a1-CreER*^T mice hip tissue. The Wnt signaling pathway has been implicated in OA in both clinical data and animal models (*Gough, 2011*), through its role in bone formation, bone growth and repair, endochondral ossification, and joint development (*Cheng et al., 2022*). In addition, growing evidence supports an important role of dysregulated Wnt signaling in chronic inflammatory diseases (*Jridi et al., 2020*). Wnt/β-catenin pathway inhibitors DKK1, Axin2, and alternative Wnt ligand Wnt5a, were highly expressed in human OA samples (*He et al., 2018*; *Ray et al., 2017*; *Martineau et al., 2017*). Meanwhile, Gremlin 1 (Wnt signaling antagonists), Frizzled-related protein (Wnt receptor), and DKK1 are recognized as key regulators of human articular cartilage homeostasis (*Leijten et al., 2012*). Furthermore, functional variants within the secreted frizzled-related protein 3 gene (Wnt receptor) are associated with hip OA in females (*Loughlin et al., 2004*). These data indicate that Wnt signaling is closely related to human OA formation. In experimental mouse models, both repression (*Zhu et al., 2008*) and forced activation (*Zhu et al., 2009*) of β-catenin caused OA. Yet how canonical Wnt signaling is dysregulated in OA remains unclear. Our data demonstrate that Cbfβ enhances articular cartilage regeneration and repair by orchestrating multiple key signaling pathways, including Wnt/β-catenin. Our results provide new insights into how Cbfβ regulates Wnt canonical signaling during OA pathogenesis which may lead to novel therapies for the treatment of degenerative joint diseases.

Our study supports that Cbfβ promotes the Hippo/Yap pathway in chondrocyte homeostasis. Yap has been reported to upregulate chondroprogenitor cells proliferation and inhibit chondrocyte maturation (*Deng et al., 2016*), and Yap1 and Runx2 protein-protein interaction has been previously confirmed (*Deng et al., 2016*). In our studies, we found that in Cbfβ-deficient cartilage, Yap transcriptional target genes *Wnt5a/b* were highly decreased. Thus, Cbfβ may promote Yap expression by regulating Runx2 expression and through a potential Cbfβ/Runx2-Yap protein-protein interaction. Some research has also indicated that Wnt5a/b-Yap signaling antagonizes canonical Wnt/β-catenin signaling and decreases expression of a panel of the major β-catenin/Tcf target genes (*Tao et al., 2017*). However, further study is needed. Our results demonstrate that the high expression level of Cbfβ in cartilage suppresses Wnt/β-catenin and low Cbfβ may lead the OA phenotype with high β-catenin expression seen in aging mice.

Tgfβ signaling also plays key roles in the development of the spontaneous OA phenotype as shown by our data. Maintaining homeostasis in articular cartilage and subchondral bone requires precise control of the Tgfβ signaling pathway (*Zhen et al., 2013*; *Che et al., 2023*). Tgfβ exerts both anabolic and catabolic effects on articular cartilage. Tgfβ stimulates the production of proteins such as aggrecan and type II collagen while also counteracting cartilage degradation by inflammatory cytokines such as IL-1 and TNF-α (*Finnson et al., 2012*). However, various components of the Tgfβ signaling pathway, along with Cbfβ, have been shown to decrease with age, illustrating a possible mechanism in the development of OA (*Qin et al., 2015*). Cbfβ and Runx1 have been revealed to be mediators of Tgfβ signaling, with the activation of Tgfβ signaling having been shown to increase Cbfβ and Runx1 expression and Cbfβ/Runx1 heterodimer formation, while *Cbfb* deletion attenuates Tgfβ signaling (*Che et al., 2023*). Our RNA sequencing results illustrate a similar pattern, where *Cbfb* conditional knockout resulted in concomitant reduction of *Tgfb1*

expression in cartilage cells and increased expression of Tgfβ signaling pathway repressors. The previously cited paper also reported that the disruption of Tgfβ signaling by the deletion of *Cbfb* in articular chondrocytes showed an increase in catabolic cytokines and enzymes interleukins and matrix metalloproteinases (*Che et al., 2023*). Furthermore, elevated levels of Tgfb1 in subchondral bone has been linked to the pathogenesis of OA (*Zhen et al., 2013*). Our work also indicated that *Cbfb* conditional knockout within *Cbfb^{f/f};Acan-CreER^T* after TMX induction resulted in significantly elevated Mmp13, suggesting a possible therapeutic target for the prevention or reduction in OA progression.

Another important component of Tgfβ signaling is Smad proteins, which are required to be phosphorylated in order to facilitate the transcription of bone and cartilage homeostasis mediators (*Hata and Chen, 2016*). Our RNA sequencing and western blot results demonstrated both altered expression and activation of several other SMADs and components of the Tgfβ signaling pathways such as Smad2 and Smad3. Our data showed that Tgfβ signaling (both P-Smad3 and Smad3) decreased in *Cbfb^{f/f};Col2a1-CreER^T* mice as shown in our results in *Figure 8*. These results were also confirmed by RNA-seq analysis as shown in the heatmaps in *Figure 5*. Che et al's study shows elevated p-Smad3 protein in *Cbfb* conditional knockout mice knee joint articular cartilage (*Che et al., 2023*). Such a difference might be due to various factors such as different age mice used for our study and Che et al's report. Tgfβ signaling has been reported for both protective and catabolic roles in the pathogenesis of OA (*Bush and Beier, 2013*). In fact, a study has shown that short and long stimulation of Tgfβ has completely opposite effects on the cartilage health (*Cherifi et al., 2021*). The dual and age-dependent role of Tgfβ signaling might cause the discrepancies observed in downstream regulators. Additionally, Tgfβ signaling is complex in the way that many signaling pathways can affect its signaling. For example, differed inflammatory states result in altered downstream Tgfβ signaling, which can also be an explanation for the deviance in results (*Baugé et al., 2007*). As such, the mechanism by which Cbfβ expression affects p-Smad2/3 requires further elucidation as it is unclear whether this functions by either a positive or negative feedback mechanism. Nevertheless, our study has well proved that Cbfβ has a crucial function in maintaining Tgfβ signaling and chondrocyte homeostasis.

In addition to those important pathways mentioned above, we also identified several significantly differentially upregulated/downregulated genes in *Cbfb* conditional knockout mice, through analysis of RNA-seq data. This includes decreased expression of *Fabp3*, *Nmrk2*, *Csf3r*, *Rgs9*, *Plin5*, *Rn7sk*, *Eif3j2* and increased expression of *Cyp2e1*, *Slc15a2*, *Alas2*, *Hba-a2*, *Lyve1*, *Snca*, *Serpina1b*, *Hbb-b1*, *Rsad2*, *Retn*, and *Trim10* in *Cbfb* conditional knockout mice. We here discussed the possible regulatory role of *Cbfb* in *Fabp3*, *Plin5*, and *Rsad2*. Of note, *Rsad2* is involved in immune regulation through its role in the NF-κB and JAK-STAT pathway (*Lin et al., 2013*). While *Plin5* is a negative regulator of PPAR signaling (*Miner et al., 2023*), a pathway which has been implicated in OA pathogenesis (*Sheng et al., 2023*). However, other differentially expressed genes that have not been discussed could also have a potentially important role in understanding the mechanism of Cbfβ in regulating chondrocyte homeostasis in OA pathogenesis. Those genes therefore need to be further studied. In addition, we conducted enrichment analysis of top upregulated and downregulated genes and identified that top significantly downregulated genes were associated with PPAR signaling, while upregulated genes were associated with TLR, Jak-STAT, and PI3K-Akt signaling, which are well known pathways associated with inflammation and immune response. These pathways could be interesting targets for further studies. Interestingly, examination of the expression profiles of different genes showed differential expression patterns between Cbfβ-deficient articular cartilage in mice hip samples and Cbfβ-deficient knee samples with their corresponding controls, suggesting that Cbfβ regulation could be tissue-specific as hip samples were composed of pure articular cartilage while knee joint samples contained articular cartilage and bone.

In summary, we found that *Cbfb* deletion in postnatal cartilage caused severe OA through the dysregulation of Wnt signaling pathways and overexpression of *Cbfb* protects against OA. Our study notably revealed that Cbfβ is a key transcription factor in articular cartilage homeostasis and promotes articular cartilage regeneration and repair in OA by orchestrating Hippo/Yap, Tgfβ, and Wnt/β-catenin signaling. The novel mechanism provides us with more insights into OA pathogenesis while also providing potential avenues for OA treatment and prevention.

## Materials and methods

### Generation of *Cbfb* inducible CKO mice

The *Cbfb^{f/f}* (Stock No: 008765) and *Acan-CreER^T* (Stock No: 019148) mouse lines were purchased from Jackson Laboratory. *Col2a1-CreER^T* mice line was generated and kindly provided by **Chen et al., 2007a**. *Cbfb^{f/f}* mice were crossed with either *Acan-CreER^T* or Col2a1-CreER^T mice to generate *Cbfb^{f/+}Col2a1-CreER^T* or *Cbfb^{f/+}Acan-CreER^T* mice, which were then intercrossed to obtain homozygous inducible CKO (*Cbfb^{f/f};Col2a1-CreER^T* and *Cbfb^{f/f};Acan-CreER^T*) mice. The genotypes of the mice were determined by polymerase chain reaction (PCR). Both male and female mice of each strain were randomly selected into groups of five animals each. The investigators were blinded during allocation, animal handling, and endpoint measurements. All mice were maintained in groups of five mice with singular sex/Breeding trios (1 male:2 females) under a 12 hr light–dark cycle with ad libitum access to regular food and water at the University of Alabama at Birmingham (UAB) Animal Facility. TMX (T5648, Sigma) was dissolved in vehicle-corn oil (C8267, Sigma) in the concentration of 10 mg/ml and vortexed until clear. The solution was aliquoted and stored at –20 °C. Before use, the TMX solution was warmed at 37°C for 5 min. 2-week-old (*Cbfb^{f/f}* mice and *Cbfb^{f/f};Col2a1-CreER^T* mice 8-week-old *Cbfb^{f/f};Acan-CreER^T*) mice received either TMX or corn oil by intraperitoneal (I.P.) injection continuously for 5 days (75 mg tamoxifen/kg body weight per day).

### DMM or ACLT surgery induced OA and AAV-*Cbfb* transduction

Eight-week-old C57BL/6 wild type mice of both sexes received either ACLT surgery, DMM surgery, or sham surgery on the right knee. We administrated AAV-CMV-*Cbfb* in a site-specific manner as described in a previous study but with minor modifications (**Wu et al., 2017a**). Briefly, mouse *Cbfb* cDNA (isoform 1, BC026749) was cloned into pAAV-MCS vector, which was followed by AAV transfection by the Ca2+-phosphate/DNA co-precipitation method. AAV titer was tested by the qPCR method. The right knee capsules were locally injected with 10 µl AAV-YFP or AAV-*Runx1* (titer >10^{10}/ml) three times on day 7, day 14, day 21 at the knee joint cavity, and euthanized 8 weeks or 10 weeks after surgery to obtain ACLT knee joint samples as described (**Zhang et al., 2022**). Mice were harvested for X-ray and histological analysis.

### Histology and tissue preparation

Histology and tissue preparation were performed as described previously (**Yang et al., 2013**). Briefly, mice were euthanized, skinned, and fixed in 4% paraformaldehyde overnight. Samples were then washed with water, dehydrated in 50% ethanol, 70% ethanol solution and then decalcified in 10% EDTA for 4 weeks. For paraffin sections, samples were dehydrated in ethanol, cleared in xylene, embedded in paraffin, and sectioned at 5 µm with a Leica microtome and mounted on frosted microscope slides (Med supply partners). H&E and SO staining were performed as described previously (**Chen et al., 2007b**). ALP staining and TRAP staining were performed with kits from Sigma.

### Radiography

Radiographs of inducible *Cbfb^{f/f};Col2a1-CreER^T* mice were detected by the Faxitron Model MX-20 at 26 kV in the UAB Small Animal Bone Phenotyping Core associated with the Center for Metabolic Bone Disease.

### Immunohistochemistry and Immunofluorescence analysis

The following primary antibodies were used: mouse-anti-Cbfβ (Santa Cruz Biotechnology Cat# sc-56751, RRID:AB_781871), mouse-anti-Col2α1 (Santa Cruz Biotechnology Cat# sc-52658, RRID:AB_2082344), rabbit-anti-MMP13 (Abcam Cat# ab39012, RRID:AB_776416), rabbit-anti-ADAMTS5 (Santa Cruz Biotechnology Cat# sc-83186, RRID:AB_2242253), rabbit-anti-Sox9 (Santa Cruz Biotechnology Cat# sc-20095, RRID:AB_661282), rabbit-anti-Yap (Santa Cruz Biotechnology Cat# sc-15407, RRID:AB_2273277), rabbit-anti-Dkk1 (Cell Signaling Technology Cat# 48367, RRID:AB_2799337), and mouse-anti-Active-β-catenin(Millipore Cat# 05–665, RRID:AB_309887). Imaging was done with a Leica DMLB Microscope and a Leica D3000 fluorescent microscope and were quantified by Image J software.

## Protein sample preparation

Mouse femoral hip articular cartilage or mouse knee cartilage was isolated, washed with sterile ice cold 1 x PBS twice, added with appropriate amount of 1 x SDS protein lysis buffer and protease inhibitor cocktail in 1.5 ml tube. Keeping on ice, femoral hip or knee tissue were quickly cut into small pieces using small scissors in 1.5 ml tube. Centrifugation was performed at room temperature at 16,000 rpm for 30 s. The supernatant was then transferred to a new, pre-chilled 1.5 ml centrifuge tube, discarding bone debris, and then boiled in water for 10 min and kept on ice. Samples were either used directly for western blot or stored at –80 °C.

## Western blot analysis

Proteins were loaded on SDS-PAGE and electro-transferred on nitrocellulose membranes. Immunoblotting was performed according to the manufacturer's instructions. The following primary antibodies were used: mouse-anti-Cbfβ (Santa Cruz Biotechnology Cat# sc-56751, RRID:AB_781871), rabbit-anti-MMP13 (Abcam Cat# ab39012, RRID:AB_776416), rabbit-anti-Yap (Santa Cruz Biotechnology Cat# sc-15407, RRID:AB_2273277), mouse-anti-GAPDH (Santa Cruz Biotechnology Cat# sc-365062, RRID:AB_10847862), mouse-anti-Active-β-catenin(Millipore Cat# 05–665, RRID:AB_309887), rabbit-anti-Smad3 (Cell Signaling Technology Cat# 9513, RRID:AB_2286450), and rabbit-anti-pSmad3 (Cell Signaling Technology Cat# 9520 (also 9520 S, 9520 P), RRID:AB_2193207). Secondary antibodies were goat anti-rabbit IgG-HRP (Santa Cruz Biotechnology Cat# sc-2004, RRID:AB_631746), and rabbit anti-mouse IgG-HRP (Santa Cruz Biotechnology Cat# sc-358917, RRID:AB_10989253). Quantification of Western blot area was performed by ImageJ.

## Cell line

The ATDC5 chondrocyte cell line (ECACC Cat# 99072806, RRID:CVCL_3894) was purchased from Millipore Sigma. The cell line was authenticated by chondrogenic differentiation and tested negative for mycoplasma.

## Primary chondrocyte culture and ATDC5 cell transfection

We isolated and cultured primary chondrocytes from neonatal f/f and *Cbfb^{f/f}Col2a1-Cre* mice as described (*Liao et al., 2021*). Primary mouse chondrocytes were induced for 7 days. Alcian blue staining was carried out to detect chondrocyte matrix deposition as previously described (*Tang et al., 2020a*). We used pMXs-GFP and pMXs-3xFlag-*Cbfb* (pMX-*Cbfb*) retroviral vectors to package and collect retroviruses, which infected ATDC5 (ECACC Cat# 99072806, RRID:CVCL_3894) cells to enhance the expression of *Cbfb*. The infected ATDC5 cell line cells were induced for 7 days before harvest for protein Western blot analysis.

## Published data analysis

Human patient information from OA cartilage samples came from prior work for RNA-seq of knee OA compared to normal controls (Accession# GSE114007; *Fisch et al., 2018*) and for methylation chip comparison of hip OA compared to hip fracture controls (Accession# GSE63695; *Rushton et al., 2014*). Analysis and comparison were performed using GEO2R and GEOprofiles. Statistical significance was assessed using Student's t-test. Values were considered statistically significant at p<0.05.

## RNA-sequencing analysis

Total RNA was isolated using TRIzol reagent (Invitrogen Corp., Carlsbad, CA) from hip articular cartilage or mouse knee cartilage and was submitted to Admera Health (South Plainsfield, NJ), who assessed sample quality with the Agilent Bioanalyzer and prepared the library using the NEBnext Ultra RNA - Poly-A kit. Libraries were analyzed using Illumina next generation sequencing and relative quantification was provided by Admera Health. Sequence reads were aligned to GRCm39/mm39 reference genome using STAR (v.2.7.9) and visualized using Integrative genomics viewer (igv v.2.16.2). Read counts were subjected to paired differential expression analysis using the R package DESeq2. Top GO downregulated categories were selected according to the p-values and enrichment score and illustrated as number of genes downregulated in respective category.

## Statistical analysis

The number of animals used in this study was determined in accordance with power analysis and our previous studies (*Tang et al., 2021*). In brief, our study used five mice per group per experiment. Data are presented as mean ± SD (n≥3). Statistical significance was assessed using Student's t test. Values were considered statistically significant at p<0.05. Results are representative of at least three individual experiments. Figures are representative of the data.

## Acknowledgements

This work was supported by the National Institutes of Health (AR-070135 and AG-056438 to WC, and AR075735 and AR074954 to YPL).

## Additional information

### Competing interests

Yi-Ping Li: Reviewing editor, *eLife*. The other authors declare that no competing interests exist.

### Funding

| Funder | Grant reference number | Author |
|---|---|---|
| National Institute of Arthritis and Musculoskeletal and Skin Diseases | AR070135 | Wei Chen |
| National Institute on Aging | AG056438 | Wei Chen |
| National Institute of Arthritis and Musculoskeletal and Skin Diseases | AR075735 | Yi-Ping Li |
| National Institute of Arthritis and Musculoskeletal and Skin Diseases | AR074954 | Yi-Ping Li |

The funders had no role in study design, data collection and interpretation, or the decision to submit the work for publication.

### Author contributions

Wei Chen, Yi-Ping Li, Conceptualization, Resources, Data curation, Software, Formal analysis, Supervision, Funding acquisition, Validation, Investigation, Visualization, Methodology, Writing - original draft, Project administration, Writing - review and editing; Yun Lu, Yan Zhang, Data curation, Formal analysis, Validation, Investigation, Visualization, Methodology, Writing - original draft, Writing - review and editing; Jinjin Wu, Data curation, Software, Formal analysis, Validation, Investigation, Visualization, Methodology, Writing - original draft, Writing - review and editing; Abigail McVicar, Data curation, Software, Formal analysis, Validation, Investigation, Visualization, Writing - original draft, Writing - review and editing; Yilin Chen, Formal analysis, Validation, Investigation, Visualization, Methodology, Writing - original draft, Writing - review and editing; Siyu Zhu, Data curation, Formal analysis, Investigation, Writing - original draft, Writing - review and editing; Guochun Zhu, Data curation, Formal analysis, Validation, Investigation, Visualization, Methodology, Writing - review and editing; You Lu, Formal analysis, Investigation, Writing - review and editing; Jiayang Zhang, Validation, Investigation, Visualization, Writing - review and editing; Matthew McConnell, Software, Validation, Investigation, Visualization, Writing - original draft, Writing - review and editing

### Author ORCIDs

Wei Chen ⓘ http://orcid.org/0000-0003-2143-0117
Yun Lu ⓘ http://orcid.org/0000-0002-4785-3401
Yilin Chen ⓘ http://orcid.org/0009-0000-9552-5990

Yi-Ping Li http://orcid.org/0000-0003-2188-6958

### Ethics

All animal experimentation was approved by the IACUC at the University of Alabama at Birmingham (protocol #21777) and Tulane University (protocol #1113) and was carried out according to the legal requirements of the Association for Assessment and Accreditation of the Laboratory Animal Care International, the University of Alabama at Birmingham Institutional Animal Care and Use Committee, and the Tulane University Institutional Animal Care and Use Committee. All studies follow NIH guidelines.

### Decision letter and Author response

Decision letter https://doi.org/10.7554/eLife.95640.sa1
Author response https://doi.org/10.7554/eLife.95640.sa2

---

## Additional files

### Supplementary files
• MDAR checklist

### Data availability

The RNA-seq data has been deposited in the Gene Expression Omnibus (GEO) under accession code GSE253210.

The following dataset was generated:

| Author(s) | Year | Dataset title | Dataset URL | Database and Identifier |
|---|---|---|---|---|
| Li Y, Chen W, Zhu S, McVicar A | 2024 | Cbfβ regulates Wnt/β-catenin, Hippo/Yap, and TGFβ signaling pathways in articular cartilage homeostasis and protects from ACLT surgery-induced osteoarthritis | https://www.ncbi.nlm.nih.gov/geo/query/acc.cgi?acc=GSE253210 | NCBI Gene Expression Omnibus, GSE253210 |

The following previously published datasets were used:

| Author(s) | Year | Dataset title | Dataset URL | Database and Identifier |
|---|---|---|---|---|
| Fisch KM, Gamini R, Alvarez-Garcia O, Akagi R | 2018 | Identification of transcription factors responsible for dysregulated networks in human osteoarthritis cartilage by global gene expression analysis | https://www.ncbi.nlm.nih.gov/geo/query/acc.cgi?acc=GSE114007 | NCBI Gene Expression Omnibus, GSE114007 |
| Rushton MD, Reynard LN, Barter MJ, Refaie R | 2014 | Characterization of the cartilage DNA methylome in knee and hip osteoarthritis | https://www.ncbi.nlm.nih.gov/geo/query/acc.cgi?acc=GSE63695 | NCBI Gene Expression Omnibus, GSE63695 |

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
