## [Editor Report]

This fundamental work advances our understanding of the role of Cbfβ in maintaining articular cartilage homeostasis and the underlying mechanisms. The evidence supporting the conclusion is convincing. This paper is of potential interest to skeletal biologists and orthopaedic surgeons who study the pathogenesis and the therapeutics of osteoarthritis.

---

## [Decision Letter]

**Decision letter after peer review:**

Thank you for submitting your article "Cbfβ regulates Wnt/β-catenin, Hippo/Yap, and TGFβ signaling pathways in articular cartilage homeostasis and protects from ACLT surgery-induced osteoarthritis" for consideration by *eLife*. Your article has been reviewed by 3 peer reviewers, and the evaluation has been overseen by a Reviewing Editor and Tony Yuen as the Senior Editor.

Essential revisions (for the authors):

This is a generally well-executed set of studies, the data from which are appropriate and largely support the conclusions. The manuscript would be strengthened by (1) including additional experiments as requested by the reviewers; and (2) providing a more comprehensive discussion and explanation to enhance clarity and accuracy.

*Reviewer #1 (Recommendations for the authors):*

The paper is generally excellent with an interesting scientific premise and strong scientific rigor. This paper targeted the discovery of OA pathogenesis and effective therapeutic approach to treat osteoarthritis. The authors found that the chondrocyte-specific deletion of Cbfβ in tamoxifen-induced Cbfβf/fCol2α1-CreERT mice caused a spontaneous OA phenotype with increased canonical Wnt signaling and inflammatory response, and decreased Hippo/YAP signaling and TGF-β signaling. Authors showed that ACLT surgery-induced OA decreased Cbfβ and Yap expression and increased active β-catenin expression in articular cartilage, while local AAV-mediated Cbfβ overexpression promoted Yap expression, diminished active β-catenin expression in OA lesions. The authors demonstrated that AAV-mediated Cbfβ overexpression in knee joints of mice with OA showed the significant protective effect of Cbfβ on articular cartilage in the ACLT OA mouse model. Overall, this study provided strong experimental data using loss-of-function and gain-of-function approaches, and in vitro and in vivo methods uncovered that low expression of Cbfβ that resulted in decreasing Wnt/β-catenin signaling, and increasing Hippo/Yap signaling and TGFβ/Smad2/3 signaling in OA articular cartilage may be the one of the OA causes/pathogenesis, and demonstrated that local admission of Cbfβ may rescue and protect OA, indicating that local Cbfβ overexpression could be an effective strategy for treatment of OA. The findings in this manuscript are novel, the manuscript is clearly written and the findings will make a significant impact in the field.

The authors' science and its presentation could be strengthened if they can make improvements as follows:

*Reviewer #2 (Recommendations for the authors):*

Concerns and recommendations:

1. Whether the Cbfβ ablation in chondrocytes affects AC development wasn't clearly disclosed. This info is crucial to help the readers to understand the disease model, particularly when TMX was administered in the early lifehood (2 wk age) and phenotypic characterization was conducted months later.

2. Justification for using both the Aggrecan-CreER and the Col2a1-CreER lines in this study was missing. Aggrecan-CreER and Col2a1-CreER targeting cells are somewhat different during development, so a clear rationale would help readers better comprehend the experiment design.

3. Methods used for βCatenin+ and Dkk1+ cell quantification (Figure 6 K,O) were not clearly described. By visual examination, the number of active βCatenin+ and Dkk1+ cells between corn oil and TMX-treated Cbfβf/f;aggrecan-CreER mice at 1 and 2.5 months (Figure 6 I, M) seemed comparable and didn't well match the graphs shown in Figure 6 K,O. Therefore, a clear disclosure of the quantification methods is crucial in this case.

4. It would be more convincing if the authors could demonstrate the overexpression of AAV-Cbfβ and colocalize its expression with βCatenin and Yap in AC cells.

5. P34 line 965-966: The nomenclatures of Cbfβf/fCol2α1-Cre and Cbfβf/fCol2α1-CreERT mice are inconsistent.

*Reviewer #3 (Recommendations for the authors):*

The authors demonstrated the gene Cbfβ promotes articular cartilage regeneration and repair in osteoarthritis (OA) through regulating Hippo/YAP signaling TGF-β signaling, and canonical Wnt signaling. Deletion of Cbfβ can induce the OA phenotypes including decreased articular cartilage and osteoblasts, and increased osteoclasts and subchondral bone hyperplasia. Deficiency of Cbfβ in cartilage can increase canonical Wnt signaling and decrease TGF-β and Hippo signaling. Overexpression of Cbfβ can inhibit Wnt signaling and enhance Hippo/YAP signaling in knee joints articular cartilage of ACLT-induced OA mice and protect against ACLT-induced OA. The manuscript is overall well-constructed, and the authors provided evidence to support their findings.

Here are some minor suggestions,

---

## [Author Response]

Essential revisions (for the authors):This is a generally well-executed set of studies, the data from which are appropriate and largely support the conclusions. The manuscript would be strengthened by (1) including additional experiments as requested by the reviewers; and (2) providing a more comprehensive discussion and explanation to enhance clarity and accuracy.

We thank the reviewers and editor for the positive appraisal of our manuscript. We greatly appreciate the insightful comments and critiques. In accordance with the reviewer’s suggestions, we have thoroughly revised all parts of the manuscript. We are glad that the reviewers considered our work to be of interest, and we are grateful for this opportunity to resubmit our manuscript.

Reviewer #1 (Recommendations for the authors):The paper is generally excellent with an interesting scientific premise and strong scientific rigor. This paper targeted the discovery of OA pathogenesis and effective therapeutic approach to treat osteoarthritis. The authors found that the chondrocyte-specific deletion of Cbfβ in tamoxifen-induced Cbfβf/fCol2α1-CreERT mice caused a spontaneous OA phenotype with increased canonical Wnt signaling and inflammatory response, and decreased Hippo/YAP signaling and TGF-β signaling. Authors showed that ACLT surgery-induced OA decreased Cbfβ and Yap expression and increased active β-catenin expression in articular cartilage, while local AAV-mediated Cbfβ overexpression promoted Yap expression, diminished active β-catenin expression in OA lesions. The authors demonstrated that AAV-mediated Cbfβ overexpression in knee joints of mice with OA showed the significant protective effect of Cbfβ on articular cartilage in the ACLT OA mouse model. Overall, this study provided strong experimental data using loss-of-function and gain-of-function approaches, and in vitro and in vivo methods uncovered that low expression of Cbfβ that resulted in decreasing Wnt/β-catenin signaling, and increasing Hippo/Yap signaling and TGFβ/Smad2/3 signaling in OA articular cartilage may be the one of the OA causes/pathogenesis, and demonstrated that local admission of Cbfβ may rescue and protect OA, indicating that local Cbfβ overexpression could be an effective strategy for treatment of OA. The findings in this manuscript are novel, the manuscript is clearly written and the findings will make a significant impact in the field.

We thank the reviewer for the positive appraisal of our manuscript. We greatly appreciate the insightful comments and critiques. In accordance with the reviewer’s suggestions, we have thoroughly revised all parts of the manuscript. We are glad that the reviewers considered our work to be of interest, and we are grateful for this opportunity to resubmit our manuscript.

Reviewer #2 (Recommendations for the authors):Concerns and recommendations:1. Whether the Cbfβ ablation in chondrocytes affects AC development wasn't clearly disclosed. This info is crucial to help the readers to understand the disease model, particularly when TMX was administered in the early lifehood (2 wk age) and phenotypic characterization was conducted months later.

We thank the reviewer for the insightful comment. In our previous work (Wu M et al., IJBS 2014) we used chondrocyte-specific *Cbfb*f/fCol2a1-Cre mice and demonstrated that they survived to adulthood and displayed severe skeletal malformation (i.e. CCD-like features). Notably, the Cbfβ deficiency in chondrocytes impaired not only growth plate formation but also trabeculae morphogenesis by regulating chondrocyte-regulating genes including those involved in Ihh/PTHrP negative-feedback loop. In this study we focused on OA pathogenesis and wanted to isolate the role of Cbfβ in the pathogenesis of OA and not during development, thus we used the inducible *Cbfb*f/fCol2a1-CreER conditional knockout mouse model in order to avoid the developmental effects of *Cbfb* conditional knockout.

2. Justification for using both the Aggrecan-CreER and the Col2a1-CreER lines in this study was missing. Aggrecan-CreER and Col2a1-CreER targeting cells are somewhat different during development, so a clear rationale would help readers better comprehend the experiment design.

We thank the reviewer for the insightful comments. While the Acan-creERT system, as employed in both current and previous studies, provides valuable insights into the role of Cbfβ in differentiated cartilage cells and its implications in the advanced stages of osteoarthritis, our current study also used *Cbfb*f/fCol2a1-CreERT aimed to explore the gene's function from a broader perspective. Previous study points out that Col2a1 is expressed in both early and late stage of chondrogenesis, including skeletal mesenchymal cells, perichondrium and presumptive joint cells, but Acan is expressed specifically in differentiated chondrocytes(1). However, studies shows that not only differentiated chondrocytes but also chondrocyte progenitors are involved in OA pathogenesis(2). In our current study, the Col2a1-CreERT system allowed us to investigate Cbfβ's role not only in mature chondrocytes but also in early chondroprogenitor cells, offering a comprehensive view of Cbfβ’s involvement in cartilage in osteoarthritis. Therefore, the use of the *Cbfb*f/fCol2a1-CreERT mouse mutant strain was instrumental in expanding our understanding of Cbfβ's multifaceted role in osteoarthritis, highlighting its importance not only in mature cartilage but also in the early stages of cartilage formation and differentiation. In addition to the different types of Cre used compared to our previous study, our current study also used gain-of-function approach in ACLT-induced OA disease model to understand the potential therapeutic function of Cbfβ in OA pathological condition. Adding our current findings to our previous research, we can now piece together a more complete picture of Cbfβ's role across the entire spectrum of cartilage development in osteoarthritis. We have added this information to the revised Discussion section. Please see lines 471-489

Blaney Davidson EN, van de Loo FA, van den Berg WB, van der Kraan PM. How to build an inducible cartilagespecific transgenic mouse. Arthritis Res Ther. 2014;16(3):210.

Tong L, Yu H, Huang X, Shen J, Xiao G, Chen L, et al. Current understanding of osteoarthritis pathogenesis and relevant new approaches. Bone Res. 2022;10(1):60.

3. Methods used for βCatenin+ and Dkk1+ cell quantification (Figure 6 K,O) were not clearly described. By visual examination, the number of active βCatenin+ and Dkk1+ cells between corn oil and TMX-treated Cbfβf/f;aggrecan-CreER mice at 1 and 2.5 months (Figure 6 I, M) seemed comparable and didn't well match the graphs shown in Figure 6 K,O. Therefore, a clear disclosure of the quantification methods is crucial in this case.

We thank the reviewer for the helpful critique. Our quantification was carried out using

NIH ImageJ software to count positive cells in 3 different experiments. Since βCatenin is a nuclear protein, we only count positive cells as cells which merge with DAPI. We double checked and clearly disclosed our quantification methods in the Materials and methods section.

4. It would be more convincing if the authors could demonstrate the overexpression of AAV-Cbfβ and colocalize its expression with βCatenin and Yap in AC cells.

We thank the reviewer for the excellent suggestion. We agree with the reviewer that it would be more convincing if we could demonstrate the overexpression of AAV-Cbfb and colocalize its expression with βCatenin and Yap in AC cells. AC cell morphology is distinct and easy to identify, and in the AAV-Cbfb group both Yap and βCatenin co-localize well in AC cells, thus we anticipate that β-Catenin and Yap would colocalize together in ACs. This study utilized a local AAV-mediated overexpression approach in the ACLT mouse model of OA. In order to carry out the addition experiments suggested by the reviewer, we would need to generate the disease model, prepare the AAV-*Cbfb*, then harvest and prepare the samples for the further histological analyses. This process would take many months and unfortunately due to time constraints we are unable to at this time. However, we appreciate the reviewer’s valuable suggestion and in our future studies we will utilize the reviewer’s suggestion.

5. P34 line 965-966: The nomenclatures of Cbfβf/fCol2α1-Cre and Cbfβf/fCol2α1-CreERT mice are inconsistent.

We thank the reviewer for bringing this to our attention. We have fixed the inconsistent nomenclature in the revised figure legend. Please see line 1032.

Reviewer #3 (Recommendations for the authors):The authors demonstrated the gene Cbfβ promotes articular cartilage regeneration and repair in osteoarthritis (OA) through regulating Hippo/YAP signaling TGF-β signaling, and canonical Wnt signaling. Deletion of Cbfβ can induce the OA phenotypes including decreased articular cartilage and osteoblasts, and increased osteoclasts and subchondral bone hyperplasia. Deficiency of Cbfβ in cartilage can increase canonical Wnt signaling and decrease TGF-β and Hippo signaling. Overexpression of Cbfβ can inhibit Wnt signaling and enhance Hippo/YAP signaling in knee joints articular cartilage of ACLT-induced OA mice and protect against ACLT-induced OA. The manuscript is overall well-constructed, and the authors provided evidence to support their findings.

We thank the reviewer for the positive appraisal of our manuscript. We greatly appreciate the insightful comments and critiques. In accordance with the reviewer’s suggestions, we have thoroughly revised all parts of the manuscript. We are glad that the reviewers considered our work to be of interest, and we are grateful for this opportunity to resubmit our manuscript.